# LET YOUR FEATURES TELL THE DIFFERENCES: UNDERSTANDING GRAPH CONVOLUTION BY FEATURE SPLITTING

**Yilun Zheng**[1][†], **Xiang Li**[2][†], **Sitao Luan**[3][*], **Xiaojiang Peng**[2], **Lihui Chen**[1][*]
[1]Nanyang Technological University, Centre for Info. Sciences and Systems,
[2]College of Big Data and Internet, Shenzhen University,
[3]Mila - Quebec Artificial Intelligence Institute.

## ABSTRACT

Graph Neural Networks (GNNs) have demonstrated strong capabilities in processing structured data. While traditional GNNs typically treat each feature dimension equally important during graph convolution, we raise an important question: *Is the graph convolution operation equally beneficial for each feature?* If not, the convolution operation on certain feature dimensions can possibly lead to harmful effects, even worse than convolution-free models. Therefore, it is required to distinguish convolution-favored and convolution-disfavored features. Traditional feature selection methods mainly focus on identifying informative features or reducing redundancy, but they are not suitable for structured data as they overlook graph structures. In graph community, some studies have investigated the performance of GNN with respect to node features using feature homophily metrics, which assess feature consistency across graph topology. Unfortunately, these metrics do not effectively align with GNN performance and cannot be reliably used for feature selection in GNNs. To address these limitations, we introduce a novel metric, Topological Feature Informativeness (TFI), to distinguish GNN-favored and GNN-disfavored features, where its effectiveness is validated through both theoretical analysis and empirical observations. Based on TFI, we propose a simple yet effective Graph Feature Selection (GFS) method, which processes GNN-favored and GNN-disfavored features with GNNs and non-GNN models separately. Compared to original GNNs, GFS significantly improves the extraction of useful topological information from each feature with comparable computational costs. Extensive experiments show that after applying GFS to **8** baseline and state-of-the-art (SOTA) GNN architectures across **10** datasets, **90%** of the GFS-augmented cases show significant performance boosts. Furthermore, our proposed TFI metric outperforms other feature selection methods for GFS. These results verify the effectiveness of both GFS and TFI. Additionally, we demonstrate that GFS's improvements are robust to hyperparameter tuning, highlighting its potential as a universally valid method for enhancing various GNN architectures. To facilitate reproducibility and further research, we have made our code publicly available at https://github.com/KTTRCDL/graph-feature-selection.

## 1 INTRODUCTION

Graph Neural Networks (GNNs) are widely used for processing graph-structured data, such as recommendation systems (Ong et al., 2023; Ong & Khong, 2024), social networks (Li et al., 2023a; Awasthi et al., 2023; Luan et al., 2019), telecommunication (Lu et al., 2024a) and bio-informatics (Zhang et al., 2021; Kang et al., 2022; Hua et al., 2024). Although graph convolution has been shown effective to enrich node features with topological information through message propagation, the performance gain is found to be restricted by the assumption of homophily, *i.e.,* similar nodes are more likely to be connected in a graph (McPherson et al., 2001). On the other hand, when a graph exhibits low homophily, *i.e.,* heterophily, the graph convolution operation can lead to performance

---

[*]Corresponding Author. [†] Equal contribution. Email address: yilun001@e.ntu.edu.sg, 2210413014@email.szu.edu.cn, sitao.luan@mail.mcgill.ca, xiaojiangpeng@sztu.edu.cn, elhchen@ntu.edu.sg.

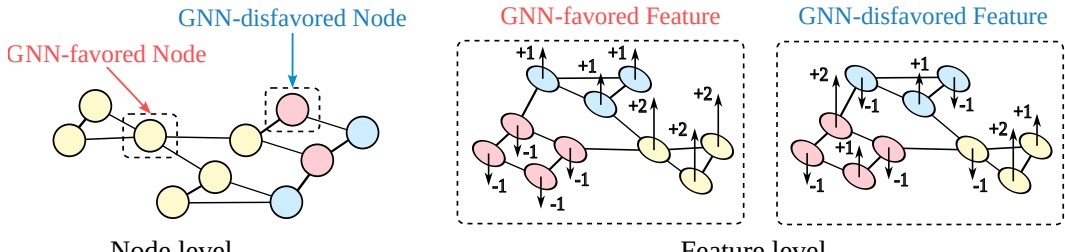

Figure 1: Improvements in GNN performance at node level and feature level. Different colors denote node labels, while the direction and magnitude of arrows denote node features.

degradation and sometimes even underperform convolution-free models, such as Multi-Layer Perceptrons (MLPs) (Zhu et al., 2020a; Luan et al., 2022b; 2024c). Therefore, to measure the impact of graph convolution operation, label-based homophily metrics (Pei et al., 2020a; Zhu et al., 2020a) are proposed to measure the label consistency along graph topology. However, they neglect the effects of node features, which is crucial for graph learning. Feature homophily metrics (Yang et al., 2021; Jin et al., 2022) are then proposed to measure the feature consistency along the graph.

Although these existing metrics can capture the feature similarity between connected nodes, they overlook that different feature dimensions may exhibit different levels of compatibility with graph structures, and thus may gain different amounts of benefits or negative impacts from graph convolution. For example, as illustrated in Figure 1 (right), the GNN-favored feature exhibits uniform values among intra-class nodes while have different values across inter-class nodes. This characteristic enables graph convolution operation to effectively increase the distinguishability among nodes from different classes, as demonstrated in (Luan et al., 2024b). Conversely, graph convolution on GNN-disfavored features may hinder the learning process. This example raises a crucial question: *How can we determine whether graph convolution is beneficial or not for a specific feature dimension?*

To address this issue, in this paper, we propose **T**opological **F**eature **I**nformativeness (**TFI**), which measures the mutual information between each dimension of aggregated node features and labels. Compared with previous feature selection metrics that mainly focus on selecting the most informative features or reducing redundancy for *i.i.d.* data (Li et al., 2017), our proposed TFI emphasizes differentiating between GNN-favored and GNN-disfavored features, which is essential for effective graph representation learning. To achieve this, we show that TFI can provably provide an upper bound on the performance gap between graph convolution and convolution-free models, which is supported by both theoretical analysis and empirical observations. Additionally, TFI overcomes the "good heterophily" issue (Ma et al., 2021; Luan et al., 2024c), which is a serious misalignment of existing feature homophily metrics and GNN performance.

Motivated by the principle of "feed the right features to the right model" Luan et al. (2022a), we propose a simple yet effective method called **G**raph **F**eature **S**election (**GFS**). GFS first uses TFI to identify GNN-favored and GNN-disfavored features. Then, to enhance the extraction of useful information, the GNN-favored features are processed by GNNs, while MLPs handle the GNN-disfavored features. Last, a final linear layer fuses the embeddings from both models to obtain the final node representation. GFS can be seamlessly integrated into almost any GNN architecture, improving overall model performance with comparable computational costs. Our experiments on real-world benchmark datasets demonstrate that GFS significantly boosts the performance of 8 GNNs across 10 datasets in node classification tasks and the improvement is robust to hyperparameter tuning. Moreover, we demonstrate that TFI outperforms other statistical and optimization-based metrics for feature selection in GNNs, validating its superiority. Besides, we surprisingly find that GFS is much more effective on node embeddings encoded by Pretrained Language Models (PLMs) than other methods. This implies that the advantages of PLMs in understanding graph-structured data might be rooted in their ability to **split the topology-aware and topology-agnostic information into separate feature dimensions**. In summary, our main contributions are as follows.

- We introduce a novel metric, Topological Feature Informativeness (TFI), to distinguish GNN-favored and GNN-disfavored features. We validate its effectiveness through both empirical observations and theoretical analysis.

- We propose Graph Feature Selection (GFS) based on TFI, a simple yet powerful method that significantly boosts GNN performance. To the best of our knowledge, we are the first to address feature selection problem based on the splitting of GNN-favored and GNN-disfavored features.
- We conduct extensive experiments by applying GFS to 8 baseline and state-of-the-art (SOTA) GNN architectures across 10 benchmark datasets. The results show a significant performance boost in **90% (72 out of 80)** of the cases.

## 2 PRELIMINARY

We define a graph as $\mathcal{G} = \{\mathcal{V}, \mathcal{E}\}$, where $\mathcal{V}$ is the node set and $\mathcal{E}$ is the edge set. For an undirected graph, its structure can be represented by an adjacency matrix $\boldsymbol{A} \in \mathbb{R}^{N \times N}$, where $N$ is the number of nodes, $A_{u,v} = A_{v,u} = 1$ indicates the presence of an edge between nodes $u$ and $v$, i.e., $e_{uv}, e_{vu} \in \mathcal{E}$, otherwise $A_{u,v} = A_{v,u} = 0$. The node classification task on graph aims to predict node labels $\boldsymbol{Y} \in \mathbb{R}^N$ by utilizing node features $\boldsymbol{X} \in \mathbb{R}^{N \times M}$ and topological information, where $X_{u,m}$ denotes the $m$-th feature of node $u$. The node degree matrix is denoted as $\boldsymbol{D} \in \mathbb{R}^{N \times N}$, where $D_u = \sum_{v \in \mathcal{V}} A_{u,v}$ is the degree of node $u$. The adjacency matrix can be normalized as $\hat{\boldsymbol{A}} = \boldsymbol{D}^{-\frac{1}{2}} \boldsymbol{A} \boldsymbol{D}^{-\frac{1}{2}}$. The neighborhood set of node $u$ are denoted by $\mathcal{N}_u = \{v | e_{uv} \in \mathcal{E}\}$.

**Graph Neural Networks.** Graph Neural Networks (GNNs), such as GCN (Kipf & Welling, 2016), GAT (Velickovié et al., 2017), and GraphSage (Hamilton et al., 2017), can effectively capture useful topological information from neighbors by message propagation mechanism. Specifically, for a node $u$, its embedding $z_u^l$ in the $l$-th layer of a GNN can be expressed as:

$$z_u^l = \text{UPDATE}\left(z_u^{l-1}, \text{AGGREGATE}(z_u^{l-1}, \{z_v^{l-1} | v \in \mathcal{N}_u\})\right) \tag{1}$$

where AGGREGATE$(\cdot)$ represents the aggregation function, and UPDATE$(\cdot)$ is the update function based on the ego node and the aggregated neighbor representations from the previous layer. For the first layer, the initial embedding is set as input node features.

The representative graph convolution can be formulated as $\boldsymbol{H}^l = \sigma(\hat{\boldsymbol{A}} \boldsymbol{H}^{l-1} \boldsymbol{W}^{l-1})$, where $\boldsymbol{H}^l$ and $\boldsymbol{W}^l$ denote the node embeddings and learnable parameter matrix at the $l$-th layer, and $\sigma(\cdot)$ is an activation function. To boost GNN performance, some architectures concatenate ego and neighborhood node embeddings as the input of the next layer (Abu-El-Haija et al., 2019; Zhu et al., 2020b), *i.e.,* $[\hat{\boldsymbol{A}} \boldsymbol{H}^{l-1}, \boldsymbol{H}^{l-1}]$ or adding the aggregated node embeddings with the ego node embeddings (Luan et al., 2022a), *i.e.,* $(\hat{\boldsymbol{A}} + \boldsymbol{I}) \boldsymbol{H}^{l-1}$, which has been shown to improve GNN performance (Platonov et al., 2023), especially in the scenario of heterophily (Luan et al., 2024c; Lu et al., 2024b).

**Label Homophily.** Homophily, a concept originating from social networks, is defined as the tendency of similar nodes are more likely to connect with each other (McPherson et al., 2001). In graph learning, people attempt to use homophily as a metric to measure whether graph convolution on a certain graph is beneficial for GNN or not. Higher homophily often implies that the topological structure can provide useful information, which can lead to better GNN performance. The commonly used label-based homophily metrics Luan et al. (2024a) are defined as follows:

$$h_{\text{edge}}(\mathcal{G}, \boldsymbol{Y}) = \frac{\left|\{e_{uv} | e_{uv} \in \mathcal{E}, Y_u = Y_v\}\right|}{|\mathcal{E}|}, \quad h_{\text{node}}(\mathcal{G}, \boldsymbol{Y}) = \frac{1}{|\mathcal{V}|} \sum_{v \in \mathcal{V}} \frac{\left|\{u | u \in \mathcal{N}_v, Y_u = Y_v\}\right|}{\left|\mathcal{N}_v\right|} \tag{2}$$

Generally, these metrics only measure label consistency across graph topology. However, they overlook the feature aspect, which also plays a critical role in GNN performance (Zheng et al., 2024a). Therefore, some studies focus on the feature consistency across graph topology.

**Feature Homophily.** To extend the conventional definition of label homophily to node features, feature homophily metrics are introduced and the general form is,

$$h(\mathcal{G}, \boldsymbol{X}_{:,m}) = \frac{1}{\eta(\boldsymbol{X}_{:,m})} \sum_{e_{uv} \in \mathcal{E}} \text{sim}(X_{u,m}, X_{v,m}) \tag{3}$$

where $\eta(\cdot)$ is a normalization function and $\text{sim}(\cdot)$ is a similarity metric. The key difference between different feature homophily metrics lies in the choice of the similarity function $\text{sim}(\cdot)$, which could be cosine similarity (Jin et al., 2022), dot-product (Yang et al., 2021), or Euclidean distance (Chen et al., 2023). Please refer to Appendix A for more detailed introduction to homophily metrics.

**Mutual Information.** Mutual Information measures the amount of information obtained about one random variable given another variable (Reza, 1994). More specifically, given variable $X$ and $Y$, the mutual information can be expressed as[1]:

$$I(X;Y) = \sum_{y \in \mathcal{Y}} \sum_{x \in \mathcal{X}} p(x,y) \, \log \frac{p(x,y)}{p(x)p(y)} \tag{4}$$

where $p(x,y)$ is joint probability, and $p(x)$ and $p(y)$ are marginal probability distributions.

## 3 TOPOLOGICAL FEATURE INFORMATIVENESS

As illustrated in Figure 1, not all features in graphs benefit from graph convolution. This raises the question: how can we determine which feature can get benefit from graph convolution? Previous studies (Zheng et al., 2024a; Luan et al., 2024b; Wang et al., 2024a) have demonstrated that it is the stable topological patterns of intra-class nodes that enable effective message aggregation from neighbors, rather than relying solely on consistency-based homophily metrics. To address the limitations of feature homophily metrics, we propose **T**opological **F**eature **I**nformativeness (**TFI**) to measure the informativeness of neighbors in graphs given the feature $\mathbf{X}_{:,\mathbf{m}}$, which is:

$$\mathrm{TFI}_m = I(\boldsymbol{Y}; \tilde{\boldsymbol{X}}_{:,\boldsymbol{m}}) \tag{5}$$

where $\tilde{\boldsymbol{X}}_{:,\boldsymbol{m}} = (\hat{\boldsymbol{A}})^k \boldsymbol{X}_{:,\boldsymbol{m}}$ is the aggregated node features from $k$-hop neighbors, $\hat{\boldsymbol{A}}$ is the adjacency matrix without self-loop, and we set $k = 1$ by default. $\mathrm{TFI}_m$ leverages mutual information to quantify the dependence between variables, including both linear and non-linear relationships.

Then, we conduct a theoretical analysis to show how $\mathrm{TFI}_m$ can effectively measure the benefit of a certain feature gained from graph convolution.

**Theorem 1.** Given a graph $\mathcal{G} = \{\mathcal{V}, \mathcal{E}\}$ with $C$ classes and $\mathrm{TFI}_m$ measured on the $m$-th feature, the prediction accuracy $P_A$ of a classifier on node labels $\boldsymbol{Y}$ with aggregated features $\tilde{\boldsymbol{X}}_{:,\boldsymbol{m}}$ is upper bounded by:

$$P_A \le \frac{\mathrm{TFI}_m + \log 2}{\log C} \tag{6}$$

The proof is given in Appendix B.

From Theorem 1, we can see that $\mathrm{TFI}_m$ can be used to estimate the effect of graph convolution on feature dimension $m$ by relating it to model performance. More specifically, a higher $\mathrm{TFI}_m$ means the prediction accuracy of a classifier with aggregated node feature $m$ has a higher upper bound, which indicates that the graph convolution operation is more likely to be beneficial for feature $m$. To guide feature selection with $\mathrm{TFI}_m$, in the following theorem, we explain how it relates to the performance gap between graph-convoluted and convolution-free features.

**Theorem 2.** The gap between $I(\boldsymbol{Y}; \tilde{\boldsymbol{X}}_{:,\boldsymbol{m}}, \boldsymbol{X}_{:,\boldsymbol{m}})$ and $I(\boldsymbol{Y}; \boldsymbol{X}_{:,\boldsymbol{m}})$ is upper bounded by $\mathrm{TFI}_m$:

$$I(\boldsymbol{Y}; \tilde{\boldsymbol{X}}_{:,\boldsymbol{m}}, \boldsymbol{X}_{:,\boldsymbol{m}}) - I(\boldsymbol{Y}; \boldsymbol{X}_{:,\boldsymbol{m}}) = I(\boldsymbol{Y}; \tilde{\boldsymbol{X}}_{:,\boldsymbol{m}}|\boldsymbol{X}_{:,\boldsymbol{m}}) \le I(\boldsymbol{Y}; \tilde{\boldsymbol{X}}_{:,\boldsymbol{m}}) = \mathrm{TFI}_m \tag{7}$$

In Theorem 2, the mutual information $I(\boldsymbol{Y}; \boldsymbol{X}_{:,\boldsymbol{m}})$ measures how well the ego node features capture the relevant information of labels, without making assumptions about the underlying data distribution. This makes $I(\boldsymbol{Y}; \boldsymbol{X}_{:,\boldsymbol{m}})$ a non-parametric estimation of MLP performance. On the other hand, $I(\boldsymbol{Y}; \tilde{\boldsymbol{X}}_{:,\boldsymbol{m}}, \boldsymbol{X}_{:,\boldsymbol{m}})$ serves as a non-parametric estimation of GNN performance, which utilizes both the ego node features $\boldsymbol{X}_{:,\boldsymbol{m}}$ and the aggregated features $\tilde{\boldsymbol{X}}_{:,\boldsymbol{m}}$ for prediction [2]. The subtraction of $I(\boldsymbol{Y}; \tilde{\boldsymbol{X}}_{:,\boldsymbol{m}}, \boldsymbol{X}_{:,\boldsymbol{m}})$ and $I(\boldsymbol{Y}; \boldsymbol{X}_{:,\boldsymbol{m}})$ indicates the information gap between graph-convoluted and convolution-free features *w.r.t.* $\boldsymbol{Y}$, which is upper bounded by $\mathrm{TFI}_m$. In other words, a feature $m$ with larger $\mathrm{TFI}_m$ means that feature $m$ is more likely to be GNN-favored; otherwise, GNN-disfavored.

To verify the effectiveness of the above claims about TFI, we conduct experiments on real-world benchmark datasets. Specifically, we first compute $\mathrm{TFI}_m$ for all features in the graph, sort them,

---

[1]For a discrete variable $Y$ and a continuous variable $X$, mutual information is estimated based on entropy using k-nearest neighbor distances, following Kraskov et al. (2004); Ross (2014).

[2]The combination of $\tilde{X}_{:,m}$ and $X_{:,m}$ could be either concatenating node embeddings, *i.e.,* $[\hat{\boldsymbol{A}}\boldsymbol{H}^{\boldsymbol{l-1}}, \boldsymbol{H}^{\boldsymbol{l-1}}]$ or adding the aggregated node embeddings with the ego node embeddings, *i.e.,* $(\hat{\boldsymbol{A}} + \boldsymbol{I})\boldsymbol{H}^{\boldsymbol{l-1}}$ as introduced in Section 2

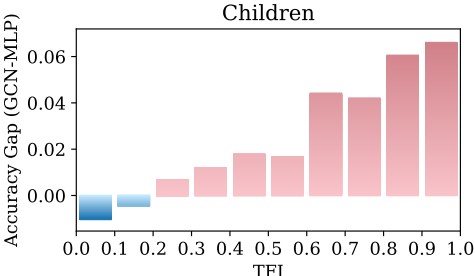 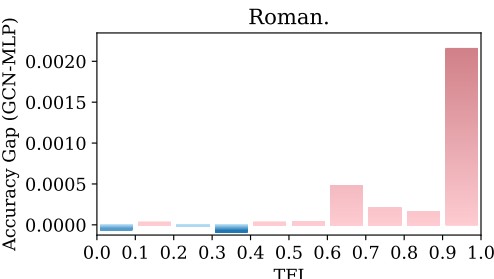

Figure 2: The performance gap between GCN and MLP with the increase of TFI.

and divide them evenly into 10 bins. Next, we train GCN and MLP only with the features in a bin respectively, and report the performance difference of GCN and MLP.

As shown in Figure 2, GCN underperforms MLP in bins with low TFI and outperforms MLP in bins with high TFI. This observation is consistent with Theorem 1 and 2, and indicates that TFI can effectively identifies GNN-favored and GNN-disfavored features. Note that the estimation of $\text{TFI}_m$ does not require all of the labels $Y$. In a semi-supervised setting, $\text{TFI}_m$ can be estimated using only the training labels and features, *i.e.*, $I(Y_{\text{train}}; \tilde{X}_{\text{train},m})$, without the need for pseudo labels.

In the next section, we will introduce how to use $\text{TFI}_m$ for feature selection in GNNs.

## 4 GRAPH FEATURE SELECTION

In this section, we propose **G**raph **F**eature **S**election (**GFS**), a TFI-based feature selection method. It is composed of three main components: (1) GNN-favored feature selection, (2) feature embedding, and (3) feature fusion.

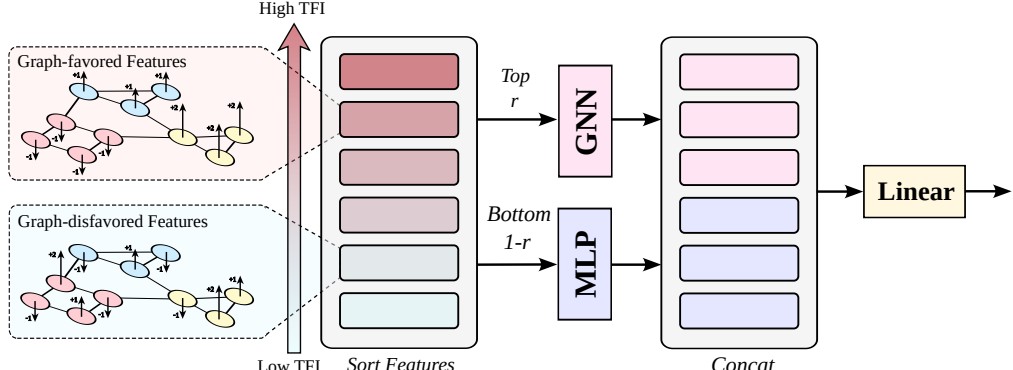

Figure 3: Framework of GFS with TFI.

As illustrated in Figure 3, for each feature dimension $m$, we first measure its corresponding $\text{TFI}_m$ using Eq. (5). Since a higher $\text{TFI}_m$ indicates stronger GNN performance compared to MLP, we apply a threshold to select the set of **GNN-favored** features as $\{X_{\mathcal{G}} | X_{:,m} \in X_{\mathcal{G}}, \text{TFI}_m \geq \delta(r)\}$, where $\delta(r)$ is the threshold for the top $r$ percentile of all TFI values in features, and $r \in (0, 1)$. The **GNN-disfavored** features are the remaining $1 - r$ of features, defined as $X_{\neg\mathcal{G}} = X \backslash \{X_{\mathcal{G}}\}$.

The GNN-favored features $X_{\mathcal{G}}$ are expected to be more suitable for GNNs, whereas the GNN-disfavored features $X_{\neg\mathcal{G}}$ are better suited for MLPs. Thus, we feed $X_{\mathcal{G}}$ into a GCN (or any other GNN architecture) and $X_{\neg\mathcal{G}}$ into an MLP to better leverage information from both neighbors and ego nodes. Specifically, in the $l$-th layer of GCN or MLP, we have:

$$\text{GCN: } H_{\mathcal{G}}^l = \sigma(\hat{A} H_{\mathcal{G}}^{l-1} W_1^{l-1}), \text{ MLP: } H_{\neg\mathcal{G}}^l = \sigma(H_{\neg\mathcal{G}}^{l-1} W_2^{l-1}) \qquad (8)$$

where $H_{\mathcal{G}}^0 = X_{\mathcal{G}}^0$ and $H_{\neg\mathcal{G}}^0 = X_{\neg\mathcal{G}}^0$. During message passing, the representations of $H_{\mathcal{G}}^l$ and $H_{\neg\mathcal{G}}^l$ are derived independently, ensuring that $X_{\mathcal{G}}^l$ and $X_{\neg\mathcal{G}}^l$ can be encoded by the most appropriate model without interference.

Lastly, to combine the node information from $X_{\mathcal{G}}$ and $X_{\neg\mathcal{G}}$ together, we concatenate the node representation from the last layer $L$ of GNN and MLP, and feed them into a linear layer:

$$H = [H_{\mathcal{G}}^L, H_{\neg\mathcal{G}}^L]W + b \tag{9}$$

where $H$ is the final node representation, $W$ is the weight matrix, and $b$ is the bias, respectively. This final node representation is then used for downstream graph tasks.

**Complexity Analysis.** To compare the time complexity of GNN+GFS against standard GNNs, we use GCN as the backbone GNN and assume the embedding size $F$ is consistent across all layers. First, estimating and sorting TFI introduces a complexity of $O(KC + N(1 + \log N))$, where $K$ is a parameter related to mutual information estimation, $C$ is the number of classes, and $N$ is the number of nodes. Both $K$ and $C$ are much smaller than $N$ or the number of edges $|\mathcal{E}|$, making this overhead negligible compared to GNN training complexity as below.

After feature selection, we obtain GNN-favored features $X_{\mathcal{G}} \in \mathbb{R}^{N \times \lceil rM \rceil}$ and GNN-disfavored features $X_{\neg\mathcal{G}} \in \mathbb{R}^{N \times \lceil (1-r)M \rceil}$, where $\lceil \cdot \rceil$ is the floor function. For the first layer of GCN+GFS, the time complexity is $O(|\mathcal{E}|\lceil rM \rceil + N \lceil rM \rceil F)$ for the GNN channel and $O(N \lceil (1-r)M \rceil F)$ for the MLP channel. Their summation, $O(|\mathcal{E}|\lceil rM \rceil + NM^2)$, is smaller than the complexity of a standard GCN, $O(|\mathcal{E}|M + NM^2)$, for $r < 1$. This is because GFS reduces the input feature dimension for GCN.

In subsequent layers, GCN+GFS has a complexity of $O(|\mathcal{E}|F + 2NF^2)$, slightly higher than the complexity of GCN $O(|\mathcal{E}|F + NF^2)$ by an additional $NF^2$ term due to the MLP layers. However, this additional cost $O(NF^2)$ is smaller than the $O(|\mathcal{E}|F)$ introduced by graph convolutions. Thus, the overall complexity of GCN+GFS is comparable to that of the original GCN.

## 5 EXPERIMENTS

To verify the effectiveness of our proposed Graph Feature Selection (GFS) and Topological Feature Informativeness (TFI), we try to answer the following questions with experiments in this section.

**RQ1**: Can GFS really enhance the performance of GNNs? Does the improvement always hold?

**RQ2**: How much does hyperparameter tuning influence the performance gains from GFS?

**RQ3**: How does TFI compare with other statistic-based and optimization-based metrics for GFS?

**RQ4**: Do the GNN-favored and GNN-disfavored features identified by TFI really fit better with GNNs or MLPs, respectively?

### 5.1 EXPERIMENTAL SETUPS

**Data Preparation.** The datasets used in our experiments include *Children, Computers, Fitness, History*, and *Photo* from Yan et al. (2023), and *Amazon-Ratings, Minesweeper, Questions, Roman-Empire*, and *Tolokers* from Platonov et al. (2023). These datasets exhibit varying levels of label homophily across different domains. The dataset statistics and descriptions are shown in Appendix C.1. For all datasets, we randomly split the data into training, validation, and test sets in a ratio of $50\% : 25\% : 25\%$ for 10 runs. Note that the node features in datasets from Yan et al. (2023) are encoded by Pretrained Language Models (PLMs).

**Baselines.** We implement GFS and compare its performance with 5 baseline GNNs, GCN (Kipf & Welling, 2016), GAT (Velicković et al., 2017), GraphSAGE (SAGE) (Hamilton et al., 2017), SGC (Wu et al., 2019) and APPNP (Gasteiger et al., 2018), and 3 SOTA GNNs, Graph Transformer (GT) (Shi et al., 2020), ACM-GCN (Luan et al., 2022a) and FAGCN (Bo et al., 2021). The GFS-augmented GNN is denoted as GNN+GFS throughout this paper. To enhance performance, we incorporate skip connections (He et al., 2016) and layer normalization (Ba et al., 2016) for all the methods, as recommended in Platonov et al. (2023); Luo et al. (2024). The detailed descriptions of these GNNs are introduced in Appendix C.2. Additionally, we compare our proposed TFI with other statistic-based metrics, including $h_{GE}$ (Jin et al., 2022), $h_{attr}$ (Yang et al., 2021), $h_{sim-cos}$ (Chen et al., 2023), $h_{sim-euc}$ (Chen et al., 2023), and $h_{CTF}$ (Lee et al., 2024). We also evaluate optimization-based metrics such as $\theta_{Soft}$ and $\theta_{Hard}$ within GFS. Detailed definitions of these statistic- and optimization-based metrics can be found in Appendix A and C.4, respectively.

**Training.** During training, we utilize the Adam optimizer (Kingma & Ba, 2014) to update model parameters. Each model is trained for 1000 epochs. Model performance is evaluated on node

classification tasks, measuring accuracy for multi-class datasets and AUC-ROC for binary-class datasets. The searching space for the ratio $r$ is $\{0.1, 0.2, \ldots, 0.9\}$ in GFS[3] and $k = 1$ for the number of the hop of neighbors in TFI[4]. See Appendix C.3 for more training details.

## 5.2 PERFORMANCE (RQ1)

The results and comparisons of the model performance are shown in Table 1, where the performance gap between GNN+GFS and GNN is denoted as $\Delta$. We observe that **(1)** GFS significantly boosts GNN performance on $90\%(72/80)$ cases of the GFS-augmented GNNs, with an average increase of $3.2\%$ in accuracy or AUC-ROC on node classification tasks with various homophily levels. This demonstrate the effectiveness of our proposed method. **(2)** We notice that the improvement of GFS varies by dataset, *e.g.,* in *Children* and *Computers*, GCN+GFS achieves a 5% average performance improvement, while in *Minesweeper* and *Questions*, the improvement is less significant. This may be because most features in *Minesweeper* and *Questions* are already GNN-favored. Nevertheless, GFS remains effective on most datasets. **(3)** We find that GFS demonstrates greater improvements on datasets encoded by Pretrained Language Models (PLMs) (40/40 cases) than those encoded by some traditional methods (32/40 cases), *e.g.,* one-hot, fasttext or statistics-based methods. This suggests that, compared to traditional feature encoding methods, which is widely used in graph learning, **PLMs potentially enable the splitting of topology-aware and topology-agnostic information into separate feature dimensions**, which helps GFS work easier. This is a surprising discovery and to our knowledge, we are the first to reveal this phenomenon. It can help explain why PLMs can assist the learning of graph models (Ye et al., 2024). In section 5.6, we will discuss more about the TFI distribution under PLM vs. traditional feature encoding methods.

Table 1: The performance of GFS on GNN baselines. The improvement is highlighted as **bold** if there is an increase, *i.e.,* $\Delta > 0$, after applying GFS.

| Model | Children | Comp. | Fitness | History | Photo | Amazon. | Mines. | Questions | Roman. | Tolokers |
|---|---|---|---|---|---|---|---|---|---|---|
| GCN | 53.88±0.37 | 83.38±1.78 | 87.56±0.79 | 84.26±0.38 | 84.48±0.65 | 51.40±0.49 | 90.01±0.53 | 76.28±1.18 | 75.51±0.65 | 84.19±0.75 |
| GCN+GFS | 59.29±0.31 | 90.23±0.18 | 93.09±0.13 | 84.99±0.38 | 87.31±0.36 | 53.32±0.81 | 89.99±0.60 | 76.50±1.38 | 83.30±0.57 | 85.01±0.91 |
| $\Delta$ | **+5.41** | **+6.85** | **+5.53** | **+0.73** | **+2.83** | **+1.92** | -0.02 | **+0.22** | **+7.79** | **+0.82** |
| GAT | 53.54±0.49 | 86.52±0.92 | 88.23±0.91 | 83.83±0.27 | 85.67±0.45 | 51.08±0.60 | 90.26±0.53 | 77.64±1.15 | 84.57±0.80 | 83.41±0.47 |
| GAT+GFS | 57.74±0.35 | 90.50±0.20 | 93.20±0.11 | 84.54±0.38 | 87.58±0.28 | 53.75±0.57 | 90.22±0.64 | 77.03±1.11 | 86.17±0.56 | 84.41±0.77 |
| $\Delta$ | **+4.20** | **+3.98** | **+4.97** | **+0.71** | **+1.91** | **+2.67** | -0.04 | -0.61 | **+1.60** | **+1.00** |
| SAGE | 54.68±0.84 | 86.08±0.50 | 88.65±1.22 | 84.06±0.42 | 85.08±0.64 | 53.80±0.56 | 90.74±0.59 | 74.91±1.06 | 82.81±0.61 | 82.77±0.38 |
| SAGE+GFS | 59.14±0.33 | 90.47±0.24 | 93.63±0.10 | 84.68±0.33 | 87.23±0.51 | 54.17±0.61 | 90.71±0.62 | 75.37±1.33 | 85.47±0.53 | 83.16±0.73 |
| $\Delta$ | **+4.46** | **+4.39** | **+4.98** | **+0.62** | **+2.15** | **+0.37** | -0.03 | **+0.46** | **+2.66** | **+0.39** |
| GT | 51.20±0.38 | 85.63±1.16 | 87.37±1.38 | 83.61±0.43 | 83.65±0.59 | 51.30±0.73 | 90.11±0.57 | 77.57±1.09 | 84.95±0.54 | 83.20±0.60 |
| GT+GFS | 56.01±0.35 | 89.97±0.27 | 92.44±0.11 | 84.09±0.32 | 87.41±0.41 | 52.47±0.54 | 89.93±0.64 | 77.70±0.73 | 86.99±0.45 | 83.46±0.67 |
| $\Delta$ | **+4.81** | **+4.34** | **+5.07** | **+0.48** | **+3.76** | **+1.17** | -0.18 | **+0.13** | **+2.04** | **+0.26** |
| SGC | 52.69±0.52 | 82.50±0.34 | 84.23±0.31 | 83.98±0.37 | 82.79±0.45 | 49.00±0.42 | 76.43±1.01 | 71.78±0.81 | 69.89±0.51 | 78.65±1.01 |
| SGC+GFS | 55.93±0.37 | 87.78±0.40 | 90.29±0.25 | 84.15±0.23 | 85.46±0.33 | 51.43±0.64 | 88.79±0.66 | 73.53±1.10 | 74.00±0.67 | 82.76±0.84 |
| $\Delta$ | **+3.24** | **+5.28** | **+6.06** | **+0.17** | **+2.67** | **+2.43** | **+12.36** | **+1.75** | **+4.11** | **+4.11** |
| APPNP | 50.63±0.89 | 83.67±0.90 | 86.76±1.18 | 83.37±0.26 | 82.20±1.41 | 48.73±0.61 | 81.61±0.79 | 75.29±1.06 | 71.48±0.65 | 79.82±1.17 |
| APPNP+GFS | 56.71±0.36 | 88.51±0.34 | 91.22±0.25 | 84.75±0.33 | 86.65±0.32 | 50.76±0.62 | 83.19±0.97 | 75.66±1.03 | 72.35±0.69 | 83.66±0.65 |
| $\Delta$ | **+6.08** | **+4.84** | **+4.46** | **+1.38** | **+4.45** | **+2.03** | **+1.58** | **+0.37** | **+0.87** | **+3.84** |
| ACMGCN | 54.60±0.50 | 85.94±0.72 | 89.10±0.98 | 84.22±0.34 | 84.99±0.34 | 51.91±0.39 | 90.59±0.58 | 76.66±1.27 | 85.27±0.57 | 83.61±0.83 |
| ACMGCN+GFS | 59.04±0.37 | 89.59±0.19 | 93.44±0.07 | 84.70±0.30 | 86.61±0.42 | 52.81±0.75 | 90.45±0.59 | 75.99±1.27 | 87.03±0.57 | 83.45±1.03 |
| $\Delta$ | **+4.44** | **+3.65** | **+4.34** | **+0.48** | **+1.62** | **+0.90** | -0.14 | -0.67 | **+1.76** | -0.16 |
| FAGCN | 50.43±0.86 | 79.92±0.99 | 83.10±0.45 | 82.04±0.62 | 80.67±0.77 | 46.08±0.52 | 78.22±2.86 | 58.60±2.08 | 62.02±2.98 | 73.07±0.70 |
| FAGCN+GFS | 56.17±0.38 | 87.72±0.49 | 89.66±0.35 | 84.32±0.36 | 85.54±0.37 | 50.72±0.82 | 88.12±1.32 | 71.94±2.03 | 72.05±1.45 | 82.82±1.28 |
| $\Delta$ | **+5.74** | **+7.80** | **+6.56** | **+2.28** | **+4.87** | **+4.64** | **+9.90** | **+13.34** | **+10.03** | **+9.75** |

## 5.3 SENSITIVITY TO HYPERPARAMETERS (RQ2)

Some recent GNNs (Li et al., 2022; Liu et al., 2022) introduce more and more hyperparameters, which could be fragile to hyperparameter tuning (Luan et al., 2024c). To investigate the sensitivity of the superiority of GFS over GNNs to ratio $r$ and other model hyperparameters, we compare the performance between GCN and GCN+GFS under varying settings. First, we test how model performance responds to the changes in ratio $r$ while fixing all other hyperparameters, and Figure 4 shows the results on 4 datasets. Note that GFS collapses to normal GCN when $r = 1.0$ and MLP when $r = 0.0$, as all features are sent to GNN or MLP, respectively. On *Children, Computer*, and *Roman-Empire*, GCN+GFS with $r$ from 0.1 to 0.9 consistently outperforms GCN ($r = 1.0$) and MLP ($r = 0.0$), indicating that the performance gain obtained from GFS is robust to hyperparameter

---

[3]To prevent the behavior of GNN+GFS from closely resembling that of traditional GNNs, we exclude $r = 1.0$ from our experiments. However, in practice, $r = 1.0$ may occur in GNN+GFS when all features are graph-favored, which would make GNN+GFS comparable to GNNs in the worst-case scenario.

[4]In Appendix D.4, we examine the influence of the number of $k$-hop neighbors on TFI. Our findings indicate that $k = 1$ is sufficient to achieve strong model performance.

$r$. We also observe that, although GCN+GFS does not significantly outperform GCN on Tolokers, sending $40\%$ ($r = 0.4$) to $90\%$ ($r = 0.9$) features to GCN is at least not worse than sending all the features into GCN ($r = 1.0$). This again demonstrates the robustness of GCN+GFS on $r$. The above results highlight that we do not always need to convolute all the features, as neither $r = 1.0$ nor $r = 0.0$ yields optimal results on most datasets.

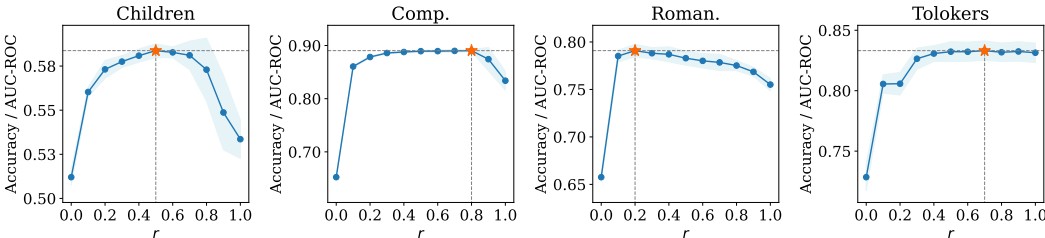

Figure 4: The impact of ratio $r$ on the performance of GCN+GFS, where $r$ features are sent to MLP and $1 - r$ features are sent to GCN. The star ★ represents the best performance.

Secondly, we analyze how other hyperparameters affect GFS performance. Figure 5 shows how GCN, MLP, and GCN+GFS respond to the changes in the number of layers, dimension of hidden embeddings, learning rate, weight decay, and dropout rate on *Computers* (homophilic graph) and *Amazon-Ratings* (heterophilic graph). Results indicate that GCN+GFS consistently outperforms GCN and MLP across various hyperparameter values. This property allows GFS to be easily integrated into most baseline GNNs without much more hyperparameter tuning. Please refer to Appendix D.2 for more results on the sensitivity of GFS to hyperparameters on the other datasets.

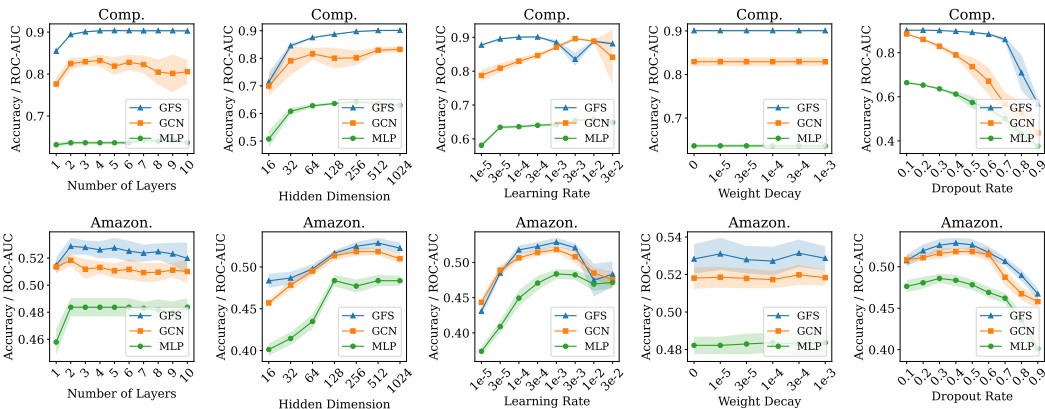

Figure 5: Comparison of the response of GCN+GFS, GCN, and MLP to 5 different hyperparameters on Computers and Amazon-Ratings.

## 5.4 EFFECTIVENESS OF TFI (RQ3)

We show the superiority of TFI over other metric-based feature selection methods on GCN+GFS, including statistic-based metrics $h_{GE}$ Jin et al. (2022), $h_{attr}$ Yang et al. (2021), $h_{sim-cos}$, $h_{sim-euc}$ Chen et al. (2023), and $h_{CTF}$ Lee et al. (2024) and optimization-based methods *e.g.*, $\theta_{Soft}$ and $\theta_{Hard}$, where feature selection process is learned in an end-to-end manner. All methods follow the same hyperparameter tuning process as in GFS. Table 2 shows the performance of different metrics in feature selection for GCN, where TFI achieves the best performance according to average rank across 10 datasets. This is reasonable because other statistic-based metrics suffer from explaining "good heterophily" issue and TFI overcomes the issue by the measurement of mutual information Platonov et al. (2024). Besides, the optimization-based metrics underperform TFI, indicating the feature selection cannot be effectively handled automatically during the optimization process of GNNs. In addition, we find that even for the metrics that are inferior to TFI, *e.g.*, $h_{attr}$, $h_{sim-euc}$, and $h_{CTF}$, applying them within our GFS framework can still improve the performance of baseline GCN, indicating the necessity of feature selection in GNNs.

Table 2: The comparison of feature selection with feature homophily metrics and optimization-based methods.

| Metrics | Children | Comp. | Fitness | History | Photo | Amazon. | Mines. | Questions | Roman. | Tolokers | Avg. Rank |
|---|---|---|---|---|---|---|---|---|---|---|---|
| TFI | **59.29±0.31** | **90.23±0.18** | **93.09±0.13** | 84.99±0.38 | **87.11±0.42** | **53.32±0.81** | 89.99±0.60 | 76.50±1.38 | **83.30±0.57** | **85.01±0.91** | **1.40** |
| $h_{GE}$ | 55.00±0.25 | 83.72±1.32 | 87.06±0.58 | 83.88±0.26 | 83.36±0.94 | 51.71±0.53 | 89.75±0.68 | 75.73±1.53 | 79.11±0.48 | 83.31±0.77 | 6.70 |
| $h_{attr}$ | 55.15±0.32 | 84.62±0.61 | 87.88±0.53 | 84.22±0.25 | 83.22±0.52 | 51.46±0.70 | 90.18±0.70 | 76.40±1.39 | 78.90±0.58 | 84.04±0.82 | 4.70 |
| $h_{sim-cos}$ | 55.15±0.51 | 84.11±0.84 | 87.09±0.98 | 84.07±0.31 | 83.38±1.01 | 51.68±0.39 | 89.87±0.53 | 75.88±1.07 | 78.98±0.66 | 83.69±0.82 | 5.95 |
| $h_{sim-euc}$ | 56.32±0.26 | 89.08±0.20 | 87.52±1.04 | **85.00±0.38** | 86.71±0.30 | 51.35±0.49 | **90.19±0.69** | **76.51±1.17** | 78.96±0.73 | 84.03±0.83 | 3.35 |
| $h_{CTF}$ | 55.04±0.51 | 84.80±0.87 | 88.69±2.25 | 84.32±0.30 | 83.46±0.85 | 51.76±0.57 | 89.97±0.68 | 76.40±1.37 | 78.70±0.38 | 83.38±0.89 | 4.55 |
| Soft | 57.04±1.37 | 85.35±3.53 | 89.54±1.32 | 84.47±0.92 | 84.83±3.28 | 51.70±0.62 | 89.93±0.60 | 75.94±1.28 | 79.20±0.46 | 84.03±0.84 | 3.50 |
| Hard | 45.28±0.79 | 66.92±4.70 | 69.48±4.14 | 80.98±0.70 | 72.09±3.72 | 42.77±0.57 | 86.62±2.25 | 75.56±1.29 | 76.96±0.61 | 79.13±0.94 | 8.90 |
| None | 53.36±1.08 | 83.38±1.78 | 87.56±0.79 | 84.13±0.29 | 84.00±0.72 | 51.40±0.49 | 89.93±0.52 | 76.24±1.48 | 75.51±0.65 | 84.19±0.75 | 5.95 |

Since TFI is not an unsupervised measurement, we investigate how the percentage of supervision in TFI (percentage of node labels used to calculate TFI in Eq. (5)) influences the model performance of GCN+GFS. As shown in Figure 6, as the supervision percentage increases, the model performance first increases and then stabilizes after 30%. This indicates that 30% of the label supervision achieves similar results as full labels in most datasets and it highlights the effectiveness of TFI in sparse label scenarios. Compared to some other methods (Zheng et al., 2024b; Li et al., 2023b) that require pseudo labels during training, TFI requires less preprocessing in semi-supervised node classification settings. Even with only 10% supervision, TFI can still enhance GCN+GFS performance on most datasets compared to the original GCN. Please refer to Appendix D.3 for more results on the influence of the supervision percentage in TFI on GFS performance.

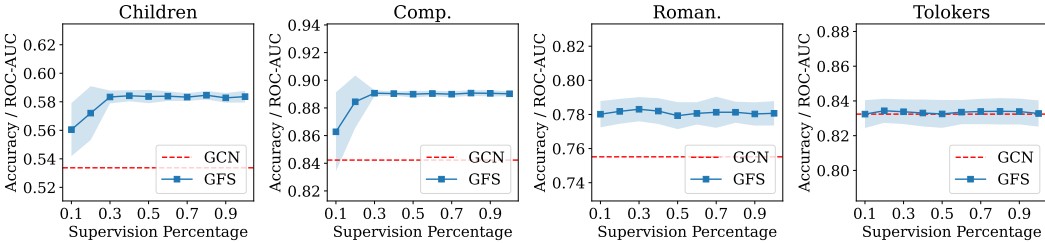

Figure 6: Influence of the supervision percentage in TFI on the model performance of GCN+GFS.

Based on TFI, we extend the feature selection from the raw node features to latent node embeddings, which may also be splitted into GNN-favored and GNN-disfavored. Specifically, we train a GCN or MLP on the training set and then get the node embeddings from the last layer. After that, we train GCN+GFS on these node embeddings following the same process as on the original node features. As shown in Figure 7, applying GCN+GFS on either the original node features, $\boldsymbol{X}$, or pretrained node embeddings, MLP($\boldsymbol{X}$) and GCN($\boldsymbol{X}, \boldsymbol{A}$), improves GCN performance on most datasets. Furthermore, for some datasets, such as *Computers, Questions*, and *Romain-Empire*, the performance of GCN+GFS could be further improved using pretrained node embeddings compared with original node features. Additionally, it would be interesting to explore the impact of graph feature selection on dynamically updated node embeddings in future research.

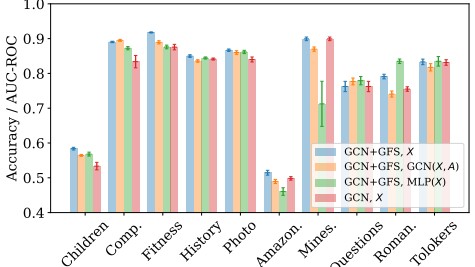

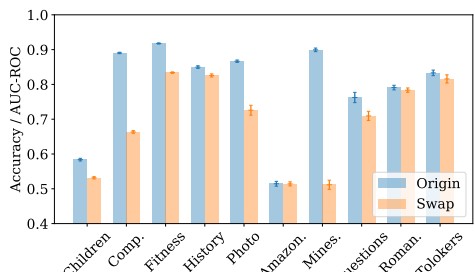

Figure 7: GCN+GFS on node features $\boldsymbol{X}$ and pretrained node embeddings GCN($\boldsymbol{X}, \boldsymbol{A}$) and MLP($\boldsymbol{X}$).

Figure 8: Performance drop of GCN+GFS by swapping GNN-favored and GNN-disfavored features.

## 5.5 GNN-FAVORED AND GNN-DISFAVORED FEATURES (RQ4)

To validate whether features selected by a high TFI are truly GNN-favored, we swapped the feature selection process by feeding GNN-favored features into MLP and GNN-disfavored features into

GNN. As shown in Figure 8, the performance of GCN+GFS drops significantly across all datasets after swapping. On some datasets, such as Minesweeper, there is a $40\%$ drop in performance. These results show that TFI can reliably identify GNN-favored and GNN-disfavored features.

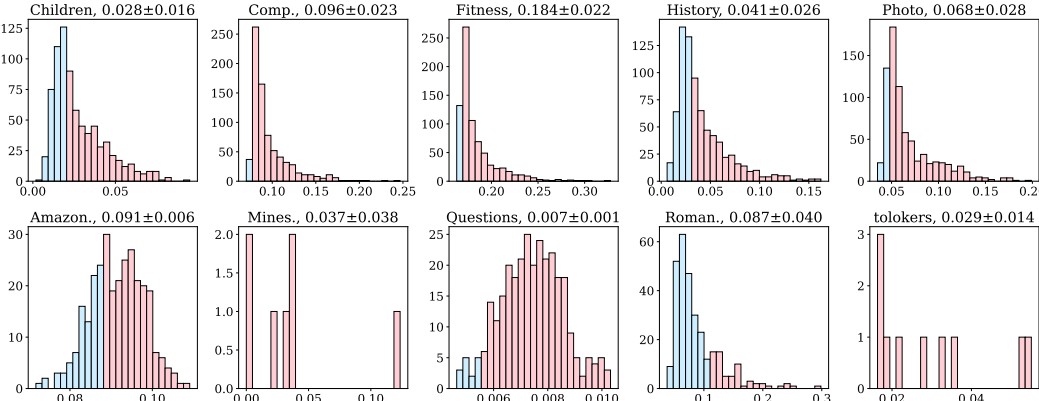

Figure 9: Comparison of histogram of TFI on 10 datasets.

## 5.6 TFI DISTRIBUTIONS: PLM VS. TRADITIONAL ENCODING METHODS

To understand why GFS is much more effective on datasets with PLM encoded features, we conduct detailed analysis on the histogram of TFI distributions for PLM (first row of Figure 9) and traditional encoding methods (second row of Figure 9). We observe that,

- **PLM:** The distributions share similar uni-modal and right-skewed shapes across different datasets. This means that most graph-disfavored feature dimensions are concentrated at the small TFI area. And the graph-favored features are long-tailed with a few feature dimensions containing the most graph-relevant information. Such distribution can help GFS efficiently split graph-aware and graph-agnostic feature dimensions and PLM can consistently produce encodings with this pattern for graph-related tasks.
- **Traditional Methods:** As shown in Appendix C.1, *Amazon-Ratings, Questions, Roman-Empire* are encoded by fasttext, Minesweeper is encoded by one-hot, and *Tolokers* is encoded by statistics-based method. Compared to PLM, these methods have lower average TFI values and show irregular TFI distributions. Although fasttext shows good performance and has an unimodal right-skewed TFI distribution on *Roman-Empire*, it fails to consistently generate high-quality features across different datasets.

## 6 CONCLUSION AND FUTURE WORKS

In this paper, we attempt to enhance GNN performance by identifying GNN-favored and GNN-disfavored features. To this end, we introduce Topological Feature Informativeness (TFI), which measures the mutual information between aggregated node features and labels. This metric effectively quantifies the performance gap between GNNs and Multi-Layer Perceptrons (MLPs), as supported by our empirical observations and theoretical analyses. We then propose a simple yet effective method, Graph Feature Selection (GFS), to incorporate TFI to improve the GNN performance. Our extensive experiments demonstrate that applying GFS to $8$ GNN architectures across $10$ datasets yields a significant performance boost in $90\%$ ($72$ out of $80$) of the cases, with an average increase of $3.2\%$ in accuracy or AUC-ROC on node classification tasks. We also show that the performance gains from GFS are robust to hyperparameter tuning, indicating its potential as a universal method for enhancing various types of GNNs.

However, the features selected by TFI introduce a new hyperparameter, ratio $r$, in GFS. Although GFS improves GNNs performance across a wide range of $r$, the best performance remains unknown due to the high complexity of the optimization process in GNNs and MLP. Therefore, it is interesting to explore the auto-selection of ratio $r$ in the future. Furthermore, since TFI is a supervised, statistic-based metric and other types of metrics may also enhance GNN performance under GFS, it would be interesting to investigate more types of metrics, both supervised and unsupervised, for feature selection of GNNs in the future.

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

## A   RELATED WORK

The proposed Graph Feature Selection (GFS) in this paper mainly relates with feature selection and GNNs efficacy. Thus, in this section, we first introduce feature selection methods on non-GNNs models and GNNs. Then, we introduce the studies on the effectiveness of GNNs from a graph level, node level, or feature level.

**Feature Selection.**   Feature selection is crucial for improving model performance, preventing overfitting, and reducing computational complexity (Li et al., 2017). To preserve data similarity, Laplacian Score (He et al., 2005) and SPEC (Zhao & Liu, 2007) are proposed to select features that best preserve the data manifold structure. To maximize the correlation between feature and class labels, MIM (Lewis, 1992) is proposed based on information theory. To minimize the fitting errors in classification tasks, Hara & Maehara (2017); Zhu et al. (2003) propose to select features with larger weights in models with $l_1$-norm regularization. To reduce feature redundancy, T-Score (Davis et al., 1986) and Chi-Square Score (Liu & Setiono, 1995) are proposed to access whether features could distinguish different classes. All of the aforementioned methods only consider data in Euclidean space, which cannot be applied for non-Euclidean data, such as graph-structured data. To address this, Ye & Liu (2012); Kim & Xing (2009) use a Graph Lasso regularizer to select features consistent across connected nodes.

While effective in traditional machine learning, these methods do not identify GNN-favored or GNN-disfavored features, as discussed in this paper. Our approach does not simply discard GNN-disfavored features; instead, it strategically utilizes them with a non-GNN model, driven by the principle of aligning the right features with the right model.

**Graph Homophily.**   Even if traditional GNNs (Kipf & Welling, 2016; Veličković et al., 2017) are believed to perform well on graph-related tasks, the performance of GNNs could be inferior than non-GNNs models in some graphs. Therefore, metrics of graph homophily, such as edge homophily (Zhu et al., 2020a), node homophily (Pei et al., 2020b), and class homophily (Lim et al., 2021), are proposed to determine when GNNs perform well. The definitions of these homophily metrics are given as:

$$h_{\text{edge}}(\mathcal{G}, \boldsymbol{Y}) = \frac{\left|\{e_{uv}|e_{uv} \in \mathcal{E}, Y_u = Y_v\}\right|}{|\mathcal{E}|}, \; h_{\text{node}}(\mathcal{G}, \boldsymbol{Y}) = \frac{1}{|\mathcal{V}|} \sum_{v \in \mathcal{V}} \frac{\left|\{u|u \in \mathcal{N}_v, Y_u = Y_v\}\right|}{\left|\mathcal{N}_v\right|} \quad (10)$$

$$h_{class}(\mathcal{G}, \boldsymbol{Y}) = \frac{1}{C-1} \sum_{c=1}^{C} \left[ \frac{\sum_{u \in \mathcal{V}, Y_u = c} \left|\{v \mid v \in \mathcal{N}_u, Y_u = Y_v\}\right|}{\sum_{u \in \{u|Y_u = c\}} d_u} - \frac{N_c}{N} \right]_+ \quad (11)$$

$$h_{adj}(\mathcal{G}, \boldsymbol{Y}) = \frac{h_{edge}(\mathcal{G}, \boldsymbol{Y}) - \sum_{c=1}^{C} \frac{D_c^2}{(2|\mathcal{E}|)^2}}{1 - \sum_{c=1}^{C} \frac{D_c^2}{(2|\mathcal{E}|)^2}} \quad (12)$$

Generally, these metrics measure the label consistency along the graph topology using an indicator function to determine if the connected nodes share the same labels. However, these metrics suffering from the "good heterophily" (Ma et al., 2021), leading to a misalignment with GNNs performance. Therefore, new metrics, such as label informativeness (Platonov et al., 2024), aggregation homophily (Luan et al., 2022a), and classifier-based homophily (Luan et al., 2024b) are proposed to mitigate this deficiency. Nevertheless, all these metrics neglect the feature aspect in graphs, which also constitute an important role in GNNs performance (Zheng et al., 2024a). Therefore, some feature homophily metrics, such as generalized edge homophily (Jin et al., 2022) is proposed to measure the feature consistency across the graph topology:

$$h_{GE}(\mathcal{G}, \boldsymbol{X}) = \frac{1}{|\mathcal{E}|} \sum_{e_{uv} \in \mathcal{E}} \frac{\boldsymbol{X}_u \boldsymbol{X}_v}{\|\boldsymbol{X}_u\| \|\boldsymbol{X}_v\|} \quad (13)$$

Similarly, local similarity (Chen et al., 2023) is proposed to measure feature homophily at the node level by cosine similarity or Euclidean similarity:

$$h_{LS-cos}(\mathcal{G}, \boldsymbol{X}) = \frac{1}{|\mathcal{V}|} \sum_{u \in \mathcal{V}} \frac{1}{d_u} \sum_{v \in \mathcal{N}_u} \frac{\boldsymbol{X}_u \boldsymbol{X}_v}{\|\boldsymbol{X}_u\| \|\boldsymbol{X}_v\|}$$

$$h_{LS-euc}(\mathcal{G}, \boldsymbol{X}) = \frac{1}{|\mathcal{V}|} \sum_{u \in \mathcal{V}} \frac{1}{d_u} \sum_{v \in \mathcal{N}_u} \left( -\|\boldsymbol{X}_u - \boldsymbol{X}_v\|_2 \right)$$

(14)

Attribute homophily (Yang et al., 2021) also consider the feature homophily but with a different normalization on each feature:

$$h_{attr,m}(\mathcal{G}, \boldsymbol{X}_{:,m}) = \frac{1}{\sum_{u \in \mathcal{V}} X_{u,m}} \sum_{u \in \mathcal{V}} \left( X_{u,m} \frac{\sum_{v \in \mathcal{N}_u} X_{v,m}}{d_u} \right)$$

$$h_{attr}(\mathcal{G}, \boldsymbol{X}) = \sum_{m=1}^{M} h_{attr,m}(\mathcal{G}, \boldsymbol{X}_{:,m})$$

(15)

Class-controlled feature homophily (Lee et al., 2024) examines the relationship between graph topology and feature dependence by analyzing the difference in expected distances between a node and its neighbors compared to random nodes, defined as follows:

$$h_{CF}(\mathcal{G}, \boldsymbol{X}, \boldsymbol{Y}) = \frac{1}{|\mathcal{V}|} \sum_{u \in \mathcal{V}} \frac{1}{d_u} \sum_{v \in \mathcal{N}_u} \Big( \boldsymbol{d}(v, \mathcal{V} \backslash \{u\}) - \boldsymbol{d}(v, \{u\}) \Big)$$

$$\boldsymbol{d}(u, \mathcal{V}') = \frac{1}{|\mathcal{V}'|} \sum_{v \in \mathcal{V}'} \|(\boldsymbol{X}_u | \boldsymbol{Y}) - (\boldsymbol{X}_v | \boldsymbol{Y})\|$$

$$\boldsymbol{X}_u | \boldsymbol{Y} = \boldsymbol{X}_u - \left( \frac{\sum_{Y_u = Y_v} \boldsymbol{X}_v}{|\{v | Y_u = Y_v, v \in \mathcal{V}\}|} \right)$$

(16)

where $\boldsymbol{X}_u | \boldsymbol{Y}$ represents class-controlled features and $\boldsymbol{d}(\cdot)$ denotes a distance function.

All of these feature homophily metrics measure the feature consistency, which still suffer from the "good heterophily" issue as similar as node homophily or edge homophily. Our proposed Topological Feature Informativeness (TFI) addresses this limitation by the mutual information, which better aligns with GNNs performance on each feature in graphs.

**Heterophily-oriented GNNs** For graphs with low homophily (heterophily), GNNs are likely to fail and even worse than non-GNNs models (Luan et al., 2024a). Therefore, many approaches (Wang et al., 2024b; Yang et al., 2021; Zheng et al., 2024b; Lu et al., 2024b; Luan et al., 2024c) are proposed to improve the performance of GNNs on heterophilous graphs, which can be categorized into graph level, node level, or feature level-based GNNs.

For the graph level, FAGCN (Bo et al., 2021) introduces high frequency signals in graph convolution to capture local information to address heterophily. Similarly, ACMGCN (Luan et al., 2022a) and FB-GNNs (Luan et al., 2022b) proposes filterbanks to fuse identify, low-pass, and high-pass filter signals. To mitigate the limitation of local heterophilous neighbors, Mixhop (Abu-El-Haija et al., 2019) and H2GCN Zhu et al. (2020b) introduces multi-hop neighbors during the graph convolution. To extend local neighbors in heterophilous graphs to global neighbors, GloGNN (Li et al., 2022) employs a signed and learnable coefficient matrix, while HiGNN (Zheng et al., 2024b) incorporates new neighbors by utilizing neighbor distribution as heterophilous information.

For the node level, SnoH (Wang et al., 2024b) modifies the message propagation for each node with varying receptive field in graph convolution. Similarly, CO-GNN (Finkelshtein et al., 2023) assigns each node with different types of directed message propagation in a cooperative manner, NoSAF (Wang et al., 2024c) provides each node with a node-specific layer aggregation with varying filter weights, and Node-MOE (Han et al., 2024) identifies heterophilous nodes with a feature inconsistency measurement, then treats these nodes differently with different filters. Different from these approaches, DisamGCL (Zhao et al., 2024) adopts a contrastive learning objective, by identifying ambiguous nodes with historical label predictions and treating these nodes as negative samples.

To our best knowledge, only DMP (Yang et al., 2021) proposes to treat each feature differently in heterophily-oriented GNNs. DMP first defines the attribute homophily as feature consistency

across the graph topology, then use a learnable weight to automatic learn the layer-wised weights for each features. However, this kind of attribute homophily suffers from "good heterophily" issue (Ma et al., 2021) and these homophily values haven't been used to guide the feature selection in graphs. Furthermore, these learnable weights in GNNs for feature selection still inferior than our statistic-based metrics, TFI.

## B  PROOF OF THEOREM 1

**Theorem 1.** Given a graph $\mathcal{G} = \{\mathcal{V}, \mathcal{E}\}$ with node features $\mathbf{X_m} \in \mathbb{R}^N$ in the $m$-th dimension and uniform node labels $\mathbf{Y} \in \mathbb{R}^N$ over $\mathcal{V}$, we can obtain aggregated features by applying graph convolution $k$ times, i.e., $\tilde{\mathbf{X}}_\mathbf{m} = (\hat{\mathbf{A}})^k \mathbf{X_m}$. For a classifier that predicts label $Y$ using the aggregated features $\tilde{X}_m$, its accuracy rate $P_A$ is upper bounded by:

$$P_A \leq \frac{I(Y; \tilde{X}_m) + \log 2}{\log(C)} \tag{17}$$

*Proof.* For a node with its aggregated feature $\tilde{X}_m$, denote its true label as $Y$ and the predicted label as $\hat{Y} = f(\tilde{X}_m)$, where $f(\cdot)$ represents the classifier. The occurrence of an error $E$ in the classifier can be expressed as:

$$E := \begin{cases} 1 & \text{if } \hat{Y} \neq Y, \\ 0 & \text{if } \hat{Y} = Y. \end{cases} \tag{18}$$

Considering the Markov chain $Y \to \tilde{X} \to \hat{Y}$, we apply Fano's inequality (Gerchinovitz et al., 2020) to obtain:

$$H(Y|\tilde{X}) \leq H_b(P_E) + P_E \log(C-1) \tag{19}$$

where $P_E$ is the error rate and $H_b(\cdot)$ is the binary entropy function.

To express $P_E$, we rearrange the inequality:

$$P_E \geq \frac{H(Y|\tilde{X}) - H_b(P_E)}{\log(C-1)} \tag{20}$$

Noting that $H(Y|\tilde{X}) = H(Y) - I(Y; \tilde{X}) = \log(C) - I(Y; \tilde{X})$ and that $H_b(P_E) \leq \log 2$, we can substitute these terms into the equation:

$$P_E \geq 1 - \frac{I(Y; \tilde{X}_m) + \log 2}{\log(C)} \tag{21}$$

Finally, converting the expression to the accuracy rate, we find:

$$P_A = 1 - P_E \leq \frac{I(Y; \tilde{X}_m) + \log 2}{\log(C)} \tag{22}$$

This concludes the proof.

## C  IMPLEMENTATION DETAILS

In this section, we provide all implementation details in Section 3 and Section 5. We have also made the code publicly available, which can be accessed here: https://github.com/KTTRCDL/graph-feature-selection

### C.1  DATASETS

We use Children, Computers, Fitness, History, Photo, Amazon-Ratings, Minesweeper, Questions, Roman-Empire, Tolokers as mentioned in Table 3 and Squirrel, Chameleon, Actor, Texas, Cornell, Wisconsin, Cora, CiteSeer, PubMed (Yang et al., 2016) as mentioned in Table 14 for all experiments. All datasets used in this work are in compliance with the MIT license. Tables 3 and 14 shows the dataset statistics and the values of homophily metrics measured on these datasets. The detailed definition of these homophily measurements are shown in Appendix A. The descriptions of main datasets are given below:

Table 3: Dataset Statistics

| Dataset | #Nodes | #Edges | #Features | #Classes | Ave. Degrees | Domain | Feat. Modeling | $h_{node}$ | $h_{edge}$ | $h_{class}$ | $h_{adj}$ | Avg. TFI |
|---|---|---|---|---|---|---|---|---|---|---|---|---|
| Children | 76,875 | 1,554,578 | 768 | 24 | 20.22 | E-commerce | PLMs | 0.4579 | 0.4220 | 0.2372 | 0.2913 | 0.0225 |
| Comp. | 87,229 | 721,081 | 768 | 10 | 8.27 | E-commerce | PLMs | 0.8469 | 0.8322 | 0.7601 | 0.7988 | 0.0208 |
| Fitness | 173,055 | 1,773,500 | 768 | 13 | 10.25 | E-commerce | PLMs | 0.8991 | 0.9004 | 0.7940 | 0.8528 | 0.0366 |
| History | 41,551 | 358,574 | 768 | 12 | 8.63 | E-commerce | PLMs | 0.7812 | 0.6626 | 0.2654 | 0.5463 | 0.0296 |
| Photo | 48,362 | 500,939 | 768 | 12 | 10.36 | E-commerce | PLMs | 0.7792 | 0.7491 | 0.7229 | 0.6892 | 0.0234 |
| Amazon. | 24,492 | 93,050 | 300 | 5 | 3.80 | E-commerce | FastText | 0.3793 | 0.3804 | 0.1270 | 0.1357 | 0.0177 |
| Mines. | 10,000 | 39,402 | 7 | 2 | 3.94 | Games | One-hot | 0.6832 | 0.6828 | 0.0094 | 0.0108 | 0.0202 |
| Questions | 48,921 | 153,540 | 301 | 2 | 3.14 | Website | FastText | 0.8963 | 0.8396 | 0.0722 | 0.2759 | 0.0049 |
| Roman. | 22,662 | 32,927 | 300 | 18 | 1.45 | Website | FastText | 0.0415 | 0.0469 | 0.0230 | -0.0778 | 0.4870 |
| tolokers | 11,758 | 519,000 | 10 | 2 | 44.14 | Social | Statistics | 0.6331 | 0.5945 | 0.1867 | 0.0887 | 0.0044 |

- Children and History (Yan et al., 2023) datasets are derived from the Amazon-Books dataset, consisting of items with the second-level label "Children" and "History" , respectively. In both datasets, nodes represent books, and edges indicate frequent co-purchases or co-views between books. Each book's label corresponds to its third-level category. The node attributes are derived from the book's title and description using Pre-trained Language Models (PLMs). The task is to classify the books into 24 categories for Children and 12 categories for History. More details can be found on: https://github.com/sktsherlock/TAG-Benchmark.

- Computers and Photo (Yan et al., 2023) datasets are extracted from the Amazon-Electronics dataset, including products with the second-level label "Computers" and "Photo", respectively. Nodes in these datasets represent electronics products, and edges signify frequent co-purchases or co-views. The labels correspond to the third-level category of the products. User reviews embedded by PLMs were used as the text attributes of the nodes, with the review having the highest number of votes being selected, or, if such a review was unavailable, a random review was chosen instead. The task is to classify the products into 10 categories for Computers and 12 categories for Photo. More details can be found on: https://huggingface.co/datasets/Sherirto/CSTAG/tree/main.

- Fitness dataset (Yan et al., 2023) is derived from the Amazon-Sports dataset, consisting of fitness-related items with the second-level label "Fitness". Nodes represent fitness products, and edges indicate frequent co-purchases or co-views between products, encoded by PLM. The labels are based on the third-level category, and the task is to classify items into 13 categories. More details can be found on: https://huggingface.co/datasets/Sherirto/CSTAG/tree/main/Fitness.

- Amazon-Ratings (Platonov et al., 2023) is derived from the Amazon product co-purchasing network metadata, provided by the SNAP (Leskovec & Krevl, 2014) Datasets, which nodes represent products and edges connect products that are frequently bought together. Node features are created using the mean of fastText (Grave et al., 2018) embeddings from product descriptions. The task is to predict the average rating of a product, grouped into five classes. More details can be found on: https://github.com/yandex-research/heterophilous-graphs/tree/main/data.

- Minesweeper (Platonov et al., 2023) is inspired by the classic Minesweeper game and is synthetic in nature. It consists of a regular 100x100 grid, where each node represents a cell connected to its eight neighboring cells (except for cells on the edges, which have fewer neighbors). In this setup, 20% of the nodes are randomly designated as mines. Each node has one-hot-encoded features representing the number of neighboring mines, with 50% of the nodes having missing features, indicated by a separate binary attribute. The task to recognize if a nodes is mine or not. More details can be found https://github.com/yandex-research/heterophilous-graphs/tree/main/data.

- Questions (Platonov et al., 2023) is based on user interactions on the Yandex Q question-answering platform. Nodes' labels represent users, and edges connect users if one answered the other's question during a one-year period. Node features are the mean of fastText (Grave et al., 2018) embeddings from user descriptions, with an additional binary feature indicating users without descriptions. The task is to predict which users remained active (i.e., not deleted or blocked) at the end of the period. More details can be found https://github.com/yandex-research/heterophilous-graphs/tree/main/data.

- Roman-Empire (Platonov et al., 2023) is constructed from the "Roman Empire" article in the English Wikipedia. Nodes in the graph represent individual words in the article, and edges connect words that either appear consecutively in the text or are related via syntactic dependencies. The node's class is syntactic role obtained using spaCy (Honnibal

et al., 2020), and fastText (Grave et al., 2018) are used for word embeddings of node features. More details can be found https://github.com/yandex-research/heterophilous-graphs/tree/main/data.

- Tolokers (Platonov et al., 2023) is based on data from the Toloka crowdsourcing platform (Likhobaba et al., 2023). The nodes represent tolokers (workers) who have participated in at least one of 13 projects and edges connect workers who have collaborated on the same task. Node features are based on profile information and task performance statistics. The task is to predict which tolokers have been banned from a project. More details can be found https://github.com/yandex-research/heterophilous-graphs/tree/main/data.

For all datasets, we randomly split the train, validation, and test set as $50\% : 25\% : 25\%$ for 10 runs. Specifically, to investigatethe impact of supervision percentage on the performance of GFS, in Figure 6, we combine the train, validation and test sets for TFI calculation with a given ratio $r$.

## C.2 MODEL

The details of all the GNN methods used in our experiments are introduced as follows:

- GCN (Kipf & Welling, 2016) performs a layer-wise propagation of node features and aggregates features from neighboring nodes to capture local graph structures. Each layer of the network updates the node embeddings by applying a convolution operation over the graph, which combines the node's own features with the features of its neighbors. The authors' implementation is available at https://github.com/tkipf/gcn.

- GAT (Velicković et al., 2017) leverages self-attention mechanisms to perform node classification on graph-structured data. The innovation of GAT lies in its ability to learn different attention coefficients for each neighboring node dynamically. Specifically, the attention coefficient between node $i$ and its neighbor $j$ is computed as: $e_{ij} = \text{LeakyReLU}(\mathbf{a}^T[W\mathbf{h}_i \| W\mathbf{h}_j])$, where $\mathbf{h}_i$ and $\mathbf{h}_j$ are the feature vectors of nodes $i$ and $j$, $W$ is a shared weight matrix, and $\mathbf{a}$ is a learnable attention vector. The coefficients are normalized using the softmax function: $\alpha_{ij} = \frac{\exp(e_{ij})}{\sum_{k \in \mathcal{N}_i} \exp(e_{ik})}$, where $\mathcal{N}_i$ denotes the set of neighbors of node $i$. The final node representation is computed as a weighted sum of its neighbors' features: $\mathbf{h}'_i = \sigma\left(\sum_{j \in \mathcal{N}_i} \alpha_{ij} W\mathbf{h}_j\right)$. The multi-head attention mechanism improves stability and expressiveness by concatenating or averaging the outputs of multiple independent attention heads, which is set with $num\_heads = 8$ in our experiments. This allows GAT to assign different importances to each neighboring node while maintaining computational efficiency. The authors' implementation is available at https://github.com/PetarV-/GAT.

- SAGE (Hamilton et al., 2017) introduces an inductive framework for node embeddings by aggregating features from a node's local neighborhood rather than requiring all nodes to be available during training. Its key innovation lies in the aggregation functions that can efficiently generate embeddings for unseen nodes. The aggregation process involves sampling and aggregating feature information from a node's neighborhood at each layer of the network. The general form of feature aggregation is: $h_v^{(k)} = \sigma\left(W^{(k)} \cdot \text{AGGREGATE}\left(h_u^{(k-1)}, \forall u \in \mathcal{N}(v)\right)\right)$, where $\mathcal{N}(v)$ represents the set of neighbors of node $v$, and AGGREGATE is set to mean operation in our experiment. This allows the model to generalize across evolving graphs and unseen nodes. The authors' implementation is available at https://github.com/williamleif/GraphSAGE.

- GT (Shi et al., 2020) is a transformer-based architecture designed for graph learning. It adapts the traditional Transformer model to graph data by incorporating node and edge features into the attention mechanism. The key innovation of GT is the use of multi-head attention to propagate node features across graph edges while also considering edge information. In our experiments, we employ the Graph Transformer with $8$ attention heads. The attention coefficients are calculated as: $\alpha_{ij} = \frac{\exp(\text{LeakyReLU}(\mathbf{q}_i^\top \mathbf{k}_j + \mathbf{e}_{ij}))}{\sum_{k \in \mathcal{N}(i)} \exp(\text{LeakyReLU}(\mathbf{q}_i^\top \mathbf{k}_k + \mathbf{e}_{ik}))}$, where $\mathbf{q}_i$ and $\mathbf{k}_j$ are the query and key vectors for nodes $i$ and $j$, and $\mathbf{e}_{ij}$ represents the edge feature between nodes $i$ and $j$. This allows the model to aggregate both node and edge information. The author's implementation is available at https://github.com/PaddlePaddle/PGL/tree/main/ogb_examples/nodeproppred/unimp

- SGC (Simplifying Graph Convolution) (Wu et al., 2019) is a simplified variant of GCN designed to reduce computational complexity while maintaining similar performance. SGC removes the non-linear activation functions between layers of a traditional GCN, collapsing multiple layers into a single linear transformation. This reduces the overall complexity of the model and results in faster training and inference times. The propagation of node features in SGC can be expressed as: $\mathbf{H} = \mathbf{S}^K \mathbf{X} \mathbf{W}$, where $\mathbf{S}$ is the normalized adjacency matrix, $\mathbf{X}$ is the input feature matrix, $\mathbf{W}$ is the weight matrix, and $K$ is the number of propagation steps (or layers). By precomputing $\mathbf{S}^K \mathbf{X}$, the model reduces to simple logistic regression on the preprocessed features, significantly speeding up the training process. In our experiments, we use $K = 1$ for optimal performance. The authors' implementation is available at https://github.com/Tiiiger/SGC.

- APPNP (Gasteiger et al., 2018) builds upon Graph Convolutional Networks (GCNs) by utilizing personalized PageRank for improved propagation of node features while avoiding oversmoothing. The model separates the neural network prediction from the propagation process, allowing it to handle larger neighborhood sizes efficiently. The propagation is controlled by the following iterative equation: $Z^{(k+1)} = (1 - \alpha)\hat{A}Z^{(k)} + \alpha H$, where $\hat{A}$ is the normalized adjacency matrix, $\alpha$ is the teleport probability (set as $\alpha = 0.1$ in our experiments), and $H$ is the initial node feature set. After $k$ (set default $k = 2$) propagation steps, the output node features are computed as: $Z = \text{softmax}(Z^{(k)})$. This structure permits the model to aggregate information from both local and distant nodes without increasing the depth of the neural network. The authors' implementation is available at https://github.com/klicperajo/ppnp.

- ACMGCN (Luan et al., 2022a) addresses the heterophily problem in graph neural networks (GNNs) by introducing an Adaptive Channel Mixing (ACM) framework. This model adapts to both homophilic and heterophilic graphs by dynamically learning to balance information from three channels: aggregation, diversification, and identity. The key innovation lies in its ability to adaptively learn different weights for each node, allowing the model to exploit local graph structure and node feature similarities. The three channels are combined using learned weights, allowing the model to emphasize different types of information for different nodes: $H^{(l)} = \alpha_L H_L^{(l)} + \alpha_H H_H^{(l)} + \alpha_I H_I^{(l)}$, where $H_L^{(l)}$, $H_H^{(l)}$, and $H_I^{(l)}$ represent the low-pass (aggregation), high-pass (diversification), and identity channels, respectively, and $\alpha_L, \alpha_H, \alpha_I$ are learned weights that balance these channels. This flexible channel mixing enables ACMGCN to significantly outperform standard GNN models on heterophilic graphs while maintaining strong performance on homophilic graphs. The authors' implementation is available at https://github.com/SitaoLuan/ACM-GNN.

- FAGCN (Bo et al., 2021) tackles the limitation of traditional GNNs that primarily focus on low-frequency signals. FAGCN introduces a self-gating mechanism to adaptively combine both low-frequency and high-frequency signals, allowing it to handle both assortative and disassortative networks effectively. The key innovation is its ability to dynamically adjust the contribution of each frequency type in the message-passing process. The aggregation of node features is expressed as: $h_i' = \epsilon h_i + \sum_{j \in \mathcal{N}(i)} \alpha_{Lij}(F_L h_j) + \alpha_{Hij}(F_H h_j)$, where $F_L$ and $F_H$ are low-pass and high-pass filters, respectively, and $\alpha_{Lij}$ and $\alpha_{Hij}$ are the learned attention coefficients for each type of signal. This enables FAGCN to adapt to different graph structures and alleviates the over-smoothing problem common in deep GNNs. The authors' implementation is available at https://github.com/bdy9527/FAGCN.

## C.3 TRAINING DETAILS

All models are implemented using the PyTorch (Ansel et al., 2024) framework and the DGL (Wang et al., 2019) library. We run experiments on a machine with 4 NVIDIA RTX A6000 GPUs, each with 48GB of memory. For most of the tables and figures illustrated in Section 5, we perform a grid search on all dataset and the hyperparameters used for search are list as follow:

- Number of layers: $\{2, 3\}$
- Hidden dimension: $\{128, 256, 512\}$
- Learning rate: $\{3 \times 10^{-5}, 10^{-4}, 3 \times 10^{-4}, 10^{-3}, 3 \times 10^{-3}, 10^{-2}\}$
- Weight decay: $\{0, 10^{-5}, 10^{-3}\}$
- Dropout rate: $\{0.1, 0.2, 0.4, 0.6, 0.8\}$

Specially, for response of GCN+GFS, GCN, and MLP to number of layers and hidden dimension, as shown in Figure 5 and Figure 12, the hyperparameters range of set are list as follow:

- Number of layers: $\{1, 2, 3, 4, 5, 6, 7, 8, 9, 10\}$
- Hidden dimension: $\{16, 32, 64, 128, 256, 512, 1024\}$
- Learning rate: $\{10^{-5}, 3 \times 10^{-5}, 10^{-4}, 3 \times 10^{-4}, 10^{-3}, 3 \times 10^{-3}, 10^{-2}, 3 \times 10^{-2}\}$
- Weight decay: $\{0, 10^{-5}, 3 \times 10^{-5}, 10^{-4}, 3 \times 10^{-4}, 10^{-3}\}$
- Dropout rate: $\{0.1, 0.2, 0.3, 0.4, 0.5, 0.6, 0.7, 0.8, 0.9\}$

To enhance the performance of all GNN models, we use skip connections (He et al., 2016) and layer normalization (Ba et al., 2016) across all methods.

Table 4: Hyperparameters for GNN baselines and GNN+GFS on Children.

| Dataset | Model | Num of Layers | Hidden Dim | Learning Rate | Weight Decay | Dropout |
|---|---|---|---|---|---|---|
| | GCN | 2 | 256 | 3e-3 | 1e-3 | 0.2 |
| | GCN+GFS | 2 | 256 | 3e-4 | 1e-4 | 0.2 |
| | GAT | 2 | 512 | 3e-5 | 0 | 0.2 |
| | GAT+GFS | 3 | 512 | 3e-4 | 0 | 0.2 |
| | SAGE | 2 | 512 | 3e-5 | 0 | 0.2 |
| | SAGE+GFS | 2 | 512 | 3e-5 | 0 | 0.2 |
| | GT | 2 | 512 | 3e-4 | 1e-3 | 0.2 |
| | GT+GFS | 3 | 512 | 3e-4 | 0 | 0.4 |
| Children | SGC | 2 | 512 | 3e-3 | 0 | 0.2 |
| | SGC+GFS | 2 | 512 | 3e-3 | 0 | 0.2 |
| | APPNP | 2 | 512 | 3e-5 | 0 | 0.2 |
| | APPNP+GFS | 2 | 512 | 3e-3 | 0 | 0.2 |
| | ACMGCN | 2 | 512 | 3e-5 | 0 | 0.2 |
| | ACMGCN+GFS | 3 | 512 | 3e-4 | 0 | 0.6 |
| | FAGCN | 2 | 512 | 3e-5 | 0 | 0.2 |
| | FAGCN+GFS | 2 | 512 | 3e-3 | 0 | 0.2 |

Table 5: Hyperparameters for GNN baselines and GNN+GFS on Computers.

| Dataset | Model | Num of Layers | Hidden Dim | Learning Rate | Weight Decay | Dropout |
|---|---|---|---|---|---|---|
| | GCN | 2 | 512 | 3e-5 | 0 | 0.2 |
| | GCN+GFS | 3 | 512 | 1e-4 | 3e-5 | 0.2 |
| | GAT | 2 | 128 | 3e-3 | 0 | 0.2 |
| | GAT+GFS | 3 | 512 | 3e-4 | 0 | 0.4 |
| | SAGE | 2 | 512 | 3e-5 | 0 | 0.2 |
| | SAGE+GFS | 3 | 512 | 3e-4 | 0 | 0.4 |
| | GT | 2 | 512 | 3e-5 | 0 | 0.2 |
| | GT+GFS | 3 | 512 | 3e-5 | 0 | 0.2 |
| Computers | SGC | 3 | 512 | 3e-4 | 0 | 0.2 |
| | SGC+GFS | 3 | 512 | 3e-3 | 0 | 0.2 |
| | APPNP | 2 | 512 | 3e-5 | 0 | 0.2 |
| | APPNP+GFS | 3 | 512 | 3e-4 | 0 | 0.2 |
| | ACMGCN | 2 | 512 | 3e-5 | 0 | 0.2 |
| | ACMGCN+GFS | 3 | 512 | 3e-4 | 0 | 0.4 |
| | FAGCN | 2 | 512 | 3e-5 | 0 | 0.2 |
| | FAGCN+GFS | 3 | 512 | 3e-3 | 0 | 0.2 |

## C.4 Optimization-based Graph Feature Selection

We implement optimization-based metrics $\theta_{\text{Soft}}$ and $\theta_{\text{Hard}}$ mentioned in Section 5.

- $\theta_{\text{Soft}}$: We use two learnable weight matrices, $\mathbf{W}_{\text{GNN}} \in \mathbb{R}^M$ and $\mathbf{W}_{\text{MLP}} \in \mathbb{R}^M$, to guide feature selection for GNN and MLP. For the input node features $\mathbf{X} \in \mathbb{R}^{N \times M}$, we divide them into a GNN-favored part $\mathbf{X}_{\mathcal{G}}$ and a GNN-disfavored part $\mathbf{X}_{\neg\mathcal{G}}$, computed as follows:

$$\mathbf{X}_{\mathcal{G}} = \frac{\mathbf{W}_{\text{GNN}}^2}{\mathbf{W}_{\text{GNN}}^2 + \mathbf{W}_{\text{MLP}}^2 + \epsilon} \cdot \mathbf{X} \tag{23}$$

$$\mathbf{X}_{\neg\mathcal{G}} = \frac{\mathbf{W}_{\text{MLP}}^2}{\mathbf{W}_{\text{GNN}}^2 + \mathbf{W}_{\text{MLP}}^2 + \epsilon} \cdot \mathbf{X} \tag{24}$$

Table 6: Hyperparameters for GNN baselines and GNN+GFS on Fitness.

| Dataset | Model | Num of Layers | Hidden Dim | Learning Rate | Weight Decay | Dropout |
|---------|-------|---------------|------------|---------------|--------------|---------|
| Fitness | GCN | 2 | 512 | 3e-5 | 0 | 0.2 |
| | GCN+GFS | 3 | 512 | 3e-4 | 0 | 0.2 |
| | GAT | 2 | 128 | 3e-3 | 0 | 0.2 |
| | GAT+GFS | 3 | 512 | 3e-4 | 0 | 0.2 |
| | SAGE | 2 | 512 | 3e-5 | 0 | 0.2 |
| | SAGE+GFS | 3 | 512 | 3e-4 | 0 | 0.2 |
| | GT | 2 | 512 | 3e-5 | 0 | 0.2 |
| | GT+GFS | 3 | 512 | 3e-4 | 0 | 0.2 |
| | SGC | 2 | 512 | 3e-3 | 0 | 0.2 |
| | SGC+GFS | 3 | 512 | 3e-3 | 0 | 0.2 |
| | APPNP | 2 | 512 | 3e-5 | 0 | 0.2 |
| | APPNP+GFS | 3 | 512 | 3e-4 | 0 | 0.6 |
| | ACMGCN | 2 | 512 | 3e-5 | 0 | 0.2 |
| | ACMGCN+GFS | 3 | 512 | 3e-4 | 0 | 0.6 |
| | FAGCN | 2 | 512 | 3e-5 | 0 | 0.2 |
| | FAGCN+GFS | 2 | 512 | 3e-3 | 0 | 0.2 |

Table 7: Hyperparameters for GNN baselines and GNN+GFS on History.

| Dataset | Model | Num of Layers | Hidden Dim | Learning Rate | Weight Decay | Dropout |
|---------|-------|---------------|------------|---------------|--------------|---------|
| History | GCN | 2 | 256 | 3e-3 | 0 | 0.4 |
| | GCN+GFS | 2 | 512 | 3e-5 | 0 | 0.2 |
| | GAT | 2 | 512 | 3e-5 | 0 | 0.2 |
| | GAT+GFS | 2 | 512 | 3e-5 | 0 | 0.2 |
| | SAGE | 2 | 512 | 3e-5 | 0 | 0.2 |
| | SAGE+GFS | 2 | 512 | 3e-5 | 0 | 0.2 |
| | GT | 2 | 512 | 3e-5 | 0 | 0.2 |
| | GT+GFS | 3 | 512 | 3e-4 | 0 | 0.2 |
| | SGC | 2 | 256 | 3e-3 | 0 | 0.2 |
| | SGC+GFS | 3 | 512 | 3e-3 | 0 | 0.2 |
| | APPNP | 2 | 512 | 3e-5 | 0 | 0.2 |
| | APPNP+GFS | 2 | 512 | 3e-3 | 0 | 0.2 |
| | ACMGCN | 2 | 512 | 3e-4 | 0 | 0.2 |
| | ACMGCN+GFS | 2 | 512 | 3e-4 | 0 | 0.6 |
| | FAGCN | 2 | 512 | 3e-5 | 0 | 0.2 |
| | FAGCN+GFS | 3 | 512 | 3e-3 | 0 | 0.2 |

Table 8: Hyperparameters for GNN baselines and GNN+GFS on Photo.

| Dataset | Model | Num of Layers | Hidden Dim | Learning Rate | Weight Decay | Dropout |
|---------|-------|---------------|------------|---------------|--------------|---------|
| Photo | GCN | 3 | 256 | 3e-3 | 0 | 0.4 |
| | GCN+GFS | 3 | 512 | 3e-5 | 0 | 0.2 |
| | GAT | 2 | 128 | 3e-3 | 0 | 0.2 |
| | GAT+GFS | 3 | 512 | 3e-4 | 0 | 0.2 |
| | SAGE | 2 | 512 | 3e-5 | 0 | 0.2 |
| | SAGE+GFS | 3 | 512 | 3e-4 | 0 | 0.2 |
| | GT | 3 | 512 | 3e-4 | 0 | 0.2 |
| | GT+GFS | 3 | 512 | 3e-5 | 0 | 0.2 |
| | SGC | 2 | 512 | 3e-3 | 1e-3 | 0.2 |
| | SGC+GFS | 3 | 512 | 3e-3 | 0 | 0.2 |
| | APPNP | 2 | 512 | 3e-5 | 0 | 0.2 |
| | APPNP+GFS | 3 | 512 | 3e-4 | 0 | 0.4 |
| | ACMGCN | 2 | 512 | 3e-3 | 0 | 0.4 |
| | ACMGCN+GFS | 3 | 512 | 3e-4 | 0 | 0.6 |
| | FAGCN | 2 | 512 | 3e-5 | 0 | 0.2 |
| | FAGCN+GFS | 3 | 512 | 3e-3 | 0 | 0.2 |

Table 9: Hyperparameters for GNN baselines and GNN+GFS on Amazon-Ratings.

| Dataset | Model | Num of Layers | Hidden Dim | Learning Rate | Weight Decay | Dropout |
|---------|-------|---------------|------------|---------------|--------------|---------|
| Amazon-Ratings | GCN | 2 | 512 | 3e-4 | 1e-3 | 0.4 |
| | GCN+GFS | 2 | 512 | 1e-3 | 1e-3 | 0.4 |
| | GAT | 2 | 512 | 3e-5 | 0 | 0.2 |
| | GAT+GFS | 3 | 512 | 3e-4 | 0 | 0.4 |
| | SAGE | 2 | 512 | 3e-5 | 0 | 0.2 |
| | SAGE+GFS | 3 | 512 | 3e-4 | 0 | 0.4 |
| | GT | 2 | 512 | 3e-5 | 0 | 0.2 |
| | GT+GFS | 3 | 512 | 3e-5 | 0 | 0.2 |
| | SGC | 3 | 512 | 3e-3 | 0 | 0.2 |
| | SGC+GFS | 2 | 512 | 3e-3 | 0 | 0.2 |
| | APPNP | 2 | 512 | 3e-5 | 0 | 0.2 |
| | APPNP+GFS | 2 | 512 | 3e-3 | 0 | 0.4 |
| | ACMGCN | 2 | 512 | 3e-4 | 0 | 0.4 |
| | ACMGCN+GFS | 3 | 512 | 3e-4 | 0 | 0.6 |
| | FAGCN | 2 | 512 | 3e-5 | 0 | 0.2 |
| | FAGCN+GFS | 2 | 512 | 3e-3 | 0 | 0.2 |

Table 10: Hyperparameters for GNN baselines and GNN+GFS on Minesweeper.

| Dataset | Model | Num of Layers | Hidden Dim | Learning Rate | Weight Decay | Dropout |
|---------|-------|---------------|------------|---------------|--------------|---------|
| Minesweeper | GCN | 2 | 512 | 3e-4 | 0 | 0.2 |
| | GCN+GFS | 2 | 128 | 1e-3 | 3e-4 | 0.2 |
| | GAT | 2 | 512 | 3e-4 | 0 | 0.2 |
| | GAT+GFS | 2 | 512 | 3e-5 | 0 | 0.2 |
| | SAGE | 2 | 512 | 3e-5 | 0 | 0.2 |
| | SAGE+GFS | 2 | 512 | 3e-5 | 0 | 0.2 |
| | GT | 2 | 512 | 3e-5 | 0 | 0.2 |
| | GT+GFS | 2 | 512 | 3e-5 | 0 | 0.2 |
| | SGC | 2 | 128 | 3e-3 | 0 | 0.2 |
| | SGC+GFS | 3 | 512 | 3e-3 | 0 | 0.2 |
| | APPNP | 2 | 512 | 3e-5 | 0 | 0.2 |
| | APPNP+GFS | 3 | 512 | 3e-4 | 0 | 0.2 |
| | ACMGCN | 3 | 512 | 3e-5 | 0 | 0.2 |
| | ACMGCN+GFS | 2 | 512 | 3e-5 | 0 | 0.2 |
| | FAGCN | 2 | 512 | 3e-5 | 0 | 0.2 |
| | FAGCN+GFS | 3 | 512 | 3e-3 | 0 | 0.2 |

Table 11: Hyperparameters for GNN baselines and GNN+GFS on Questions.

| Dataset | Model | Num of Layers | Hidden Dim | Learning Rate | Weight Decay | Dropout |
|---------|-------|---------------|------------|---------------|--------------|---------|
| Questions | GCN | 2 | 512 | 3e-4 | 0 | 0.2 |
| | GCN+GFS | 3 | 128 | 1e-4 | 1e-3 | 0.2 |
| | GAT | 2 | 512 | 3e-4 | 0 | 0.2 |
| | GAT+GFS | 2 | 512 | 3e-5 | 0 | 0.2 |
| | SAGE | 2 | 512 | 3e-5 | 0 | 0.2 |
| | SAGE+GFS | 2 | 512 | 3e-5 | 0 | 0.2 |
| | GT | 2 | 512 | 3e-5 | 0 | 0.2 |
| | GT+GFS | 3 | 512 | 3e-5 | 0 | 0.4 |
| | SGC | 3 | 512 | 3e-3 | 0 | 0.8 |
| | SGC+GFS | 3 | 512 | 3e-5 | 0 | 0.6 |
| | APPNP | 2 | 512 | 3e-5 | 0 | 0.2 |
| | APPNP+GFS | 3 | 512 | 3e-5 | 0 | 0.6 |
| | ACMGCN | 2 | 512 | 3e-5 | 0 | 0.2 |
| | ACMGCN+GFS | 2 | 512 | 3e-5 | 0 | 0.2 |
| | FAGCN | 2 | 512 | 3e-5 | 0 | 0.2 |
| | FAGCN+GFS | 3 | 512 | 3e-3 | 0 | 0.2 |

Table 12: Hyperparameters for GNN baselines and GNN+GFS on Roman-Empire.

| Dataset | Model | Num of Layers | Hidden Dim | Learning Rate | Weight Decay | Dropout |
|---|---|---|---|---|---|---|
| Roman-Empire | GCN | 2 | 512 | 3e-5 | 0 | 0.2 |
| | GCN+GFS | 3 | 128 | 1e-2 | 1e-4 | 0.3 |
| | GAT | 2 | 128 | 3e-3 | 0 | 0.4 |
| | GAT+GFS | 3 | 512 | 3e-4 | 0 | 0.6 |
| | SAGE | 2 | 512 | 3e-3 | 0 | 0.6 |
| | SAGE+GFS | 3 | 512 | 3e-4 | 0 | 0.4 |
| | GT | 2 | 512 | 3e-3 | 0 | 0.8 |
| | GT+GFS | 3 | 512 | 3e-4 | 0 | 0.6 |
| | SGC | 2 | 128 | 3e-3 | 0 | 0.4 |
| | SGC+GFS | 2 | 512 | 3e-3 | 0 | 0.2 |
| | APPNP | 2 | 512 | 3e-5 | 0 | 0.2 |
| | APPNP+GFS | 2 | 512 | 3e-5 | 0 | 0.2 |
| | ACMGCN | 2 | 512 | 3e-3 | 0 | 0.8 |
| | ACMGCN+GFS | 3 | 512 | 3e-4 | 0 | 0.8 |
| | FAGCN | 2 | 512 | 3e-5 | 0 | 0.2 |
| | FAGCN+GFS | 2 | 512 | 3e-3 | 0 | 0.2 |

Table 13: Hyperparameters for GNN baselines and GNN+GFS on Tolokers.

| Dataset | Model | Num of Layers | Hidden Dim | Learning Rate | Weight Decay | Dropout |
|---|---|---|---|---|---|---|
| Tolokers | GCN | 2 | 512 | 3e-4 | 1e-5 | 0.2 |
| | GCN+GFS | 3 | 128 | 1e-3 | 3e-5 | 0.2 |
| | GAT | 2 | 512 | 3e-5 | 0 | 0.2 |
| | GAT+GFS | 3 | 512 | 3e-5 | 0 | 0.2 |
| | SAGE | 2 | 512 | 3e-5 | 0 | 0.2 |
| | SAGE+GFS | 3 | 512 | 3e-3 | 0 | 0.4 |
| | GT | 3 | 512 | 3e-5 | 0 | 0.2 |
| | GT+GFS | 3 | 512 | 3e-5 | 0 | 0.2 |
| | SGC | 2 | 512 | 3e-3 | 0 | 0.2 |
| | SGC+GFS | 3 | 512 | 3e-3 | 0 | 0.2 |
| | APPNP | 2 | 512 | 3e-5 | 0 | 0.2 |
| | APPNP+GFS | 3 | 512 | 3e-3 | 0 | 0.2 |
| | ACMGCN | 3 | 512 | 3e-4 | 0 | 0.2 |
| | ACMGCN+GFS | 3 | 512 | 3e-4 | 0 | 0.2 |
| | FAGCN | 2 | 512 | 3e-5 | 0 | 0.2 |
| | FAGCN+GFS | 3 | 512 | 3e-3 | 0 | 0.2 |

where $\epsilon$ is set to $10^{-7}$ to prevent division by zero. In this formulation, $\mathbf{W}_{\text{GNN}}$ represents the proportion of each feature that is GNN-favored, while $\mathbf{W}_{\text{MLP}}$ represents the proportion that is MLP-favored. The sum of $\mathbf{X}_{\mathcal{G}}$ and $\mathbf{X}_{\neg\mathcal{G}}$ equals the original input $\mathbf{X}$, indicating that the features are divided based on the relative influence of GNN and MLP preferences. These weight parameters are updated during the training process.

- $\theta_{\text{Hard}}$: We use a learnable weight matrix, $\mathbf{W}_{\text{Hard}} \in \mathbb{R}^{M \times 2}$, to guide feature selection for GNN and MLP. For the input node features $\mathbf{X} \in \mathbb{R}^{N \times M}$, we divide them into a GNN-favored part $\mathbf{X}_{\mathcal{G}}$ and a GNN-disfavored part $\mathbf{X}_{\neg\mathcal{G}}$, computed as follows:

$$\mathbf{M}_{\text{Hard}} = \text{Gumbel-Softmax}(\mathbf{W}_{\text{Hard}}) \tag{25}$$

$$\mathbf{X}_{\mathcal{G}} = \mathbf{X} \cdot \mathbf{M}_{\text{Hard}}[:, 0] \tag{26}$$

$$\mathbf{X}_{\neg\mathcal{G}} = \mathbf{X} \cdot \mathbf{M}_{\text{Hard}}[:, 1] \tag{27}$$

In this case, $\mathbf{M}_{\text{Hard}} \in \mathbb{R}^{M \times 2}$ is the output of the Gumbel-Softmax applied to $\mathbf{W}_{\text{Hard}}$. For each row in $\mathbf{M}_{\text{Hard}}$, one value will be 1 and the other will be 0. A value of 1 indicates the selection of that feature, meaning that either the GNN-favored part or the MLP-favored part is chosen for each feature. This binary selection mechanism ensures that each feature is exclusively assigned to either $\mathbf{X}_{\mathcal{G}}$ or $\mathbf{X}_{\neg\mathcal{G}}$, depending on which component is favored. The use of Gumbel-Softmax($\cdot$) (Jang et al., 2017) allows for differentiable sampling, enabling the weight parameters to be updated during the training process.

## C.5 DETAILS OF PRETRAINED NODE EMBEDDINGS

We implement experiments using both GCN and MLP pretrained features to evaluate their effect when applying GCN+GFS on either the original node features, $X$, or on the pretrained node embeddings, MLP($X$) and GCN($X, A$). The results of this evaluation are presented in Figure 7 in the main text.

- MLP($X$): We pretrain a Multi-Layer Perceptron (MLP) model with the following setup: 2 layers, a hidden dimension of 128, trained for 1000 steps using the ELU activation function. For each of the 10 data splits, the MLP is trained separately with the Adam optimizer, using a learning rate of $1e - 2$ and a weight decay of $5e - 4$. The pretrained MLP is then used to extract MLP($X$) corresponding to the training splits, which serves as the node embeddings for the experiment.

- GCN($X, A$): We pretrain a Graph Convolutional Network (GCN) with 2 layers, a hidden dimension of 128, and trained for 1000 steps. The activation function used is ReLU, with a dropout rate of 0.5. Similarly, GCN is trained separately on each of the 10 data splits using the Adam optimizer, with a learning rate of $1e - 2$ and a weight decay of $5e - 4$. The pretrained GCN is then used to extract GCN($X, A$), corresponding to the training splits, as the node embeddings.

# D    MORE EXPERIMENTAL RESULTS

## D.1    GCN+GFS ON ADDITIONAL DATASETS

We report the performance of GCN+GFS on additional datasets including Squirrel, Chameleon, Actor, Texas, Cornell, Wisconsin, Cora, CiteSeer, and PubMed. The dataset statistics and descriptions are shown in Figure 14.

Table 14: Addtional Dataset Statistics

| Dataset | #Nodes | #Edges | #Features | #Classes | Ave. Degrees | $h_{node}$ | $h_{edge}$ | $h_{class}$ | $h_{adj}$ | Avg. TFI |
|---------|--------|--------|-----------|----------|--------------|------------|------------|-------------|-----------|----------|
| Squirrel | 2223 | 46998 | 2089 | 5 | 21.14 | 0.1759 | 0.2072 | 0.0725 | -0.0076 | 0.0173 |
| Chameleon | 890 | 8854 | 2325 | 5 | 9.95 | 0.2273 | 0.2361 | 0.0601 | 0.0360 | 0.0177 |
| Actor | 7600 | 26659 | 932 | 5 | 3.51 | 0.2197 | 0.2167 | 0.0074 | 0.0015 | 0.0029 |
| Texas | 183 | 279 | 1703 | 5 | 1.52 | 0.0748 | 0.0609 | 0.0017 | -1.4628 | 0.0348 |
| Cornell | 183 | 277 | 1703 | 5 | 1.51 | 0.1246 | 0.1227 | 0.0482 | -1.1984 | 0.0202 |
| Wisconsin | 251 | 450 | 1703 | 5 | 1.79 | 0.1934 | 0.1778 | 0.0447 | -0.3947 | 0.0182 |
| Cora | 2708 | 10556 | 1433 | 7 | 3.90 | 0.8252 | 0.8100 | 0.7657 | 0.7717 | 0.0188 |
| CiteSeer | 3327 | 9228 | 3703 | 6 | 2.77 | 0.7166 | 0.7391 | 0.6267 | 0.6673 | 0.0103 |
| PubMed | 19717 | 88651 | 500 | 3 | 4.50 | 0.7924 | 0.8024 | 0.6641 | 0.6836 | 0.0440 |

As shown in Figure 10, the performance of the GCN+GFS model surpasses the standard GCN across additional datasets. The selective feature processing approach provided by GFS allows GNNs to focus on beneficial features, leading to improved performance across diverse datasets.

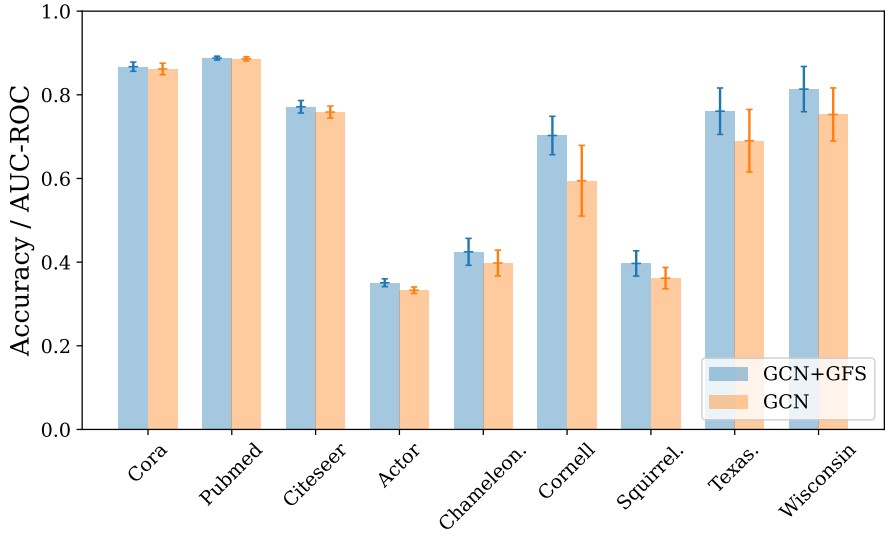

Figure 10: Performance on additional datasets.

## D.2    SENSITIVITY OF GFS

Figure 11 shows the response of GCN+GFS to $r$ on 6 other datasets, where GFS collapses to a GNN when $r = 1.0$ or an MLP when $r = 0.0$, as all features are sent to GNN or MLP, respectively. Although in some datasets like Minesweeper and Questions, where all the features may favor GNNs, GFS doesn't significantly improve GCN performance, it doesn't diminish GFS's overall effectiveness on most datasets.

Figure 12 shows how GCN, MLP, and GCN+GFS respond to changes in number of layers, dimension of hidden embeddings, learning rate, weight decay, and dropout rate on a homophilous graph (Computers) and a heterophilous graph (Amazon-Ratings) on other 6 datasets. In most cases, GFS outperforms GCN, but for certain hyperparameter settings on the Questions dataset, GFS does not consistently outperform GCN. This is likely because the node features in these datasets are already highly graph-favored, making the additional feature selection offered by GFS less impactful. These results highlight the robustness of GFS under different hyperparameter settings.

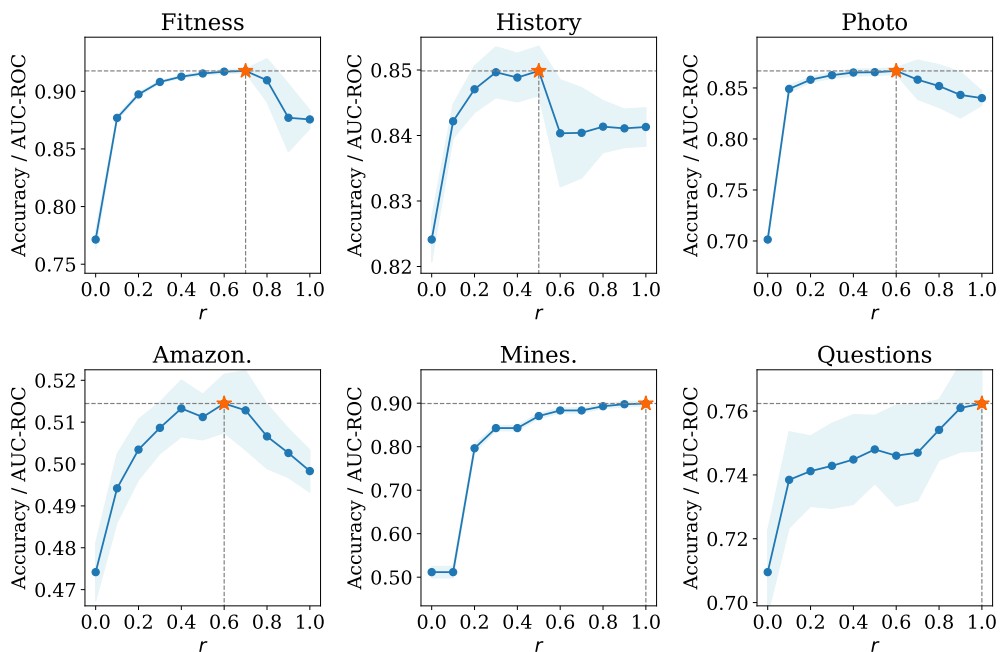

Figure 11: The performance of GCN+GFS is shown as the ratio $r$ of GNN-favored features in TFI increases. The point representing the best performance is highlighted as ★.

### D.3  PERCENTAGE OF SUPERVISION IN TFI

In Figure 13, the influence of the supervision percentage in TFI on the performance of GCN+GFS across other 6 datasets is shown. The results highlight that as the supervision percentage increases, the performance of GCN+GFS generally improves or stabilizes. Even with only $10\%$ supervision, GCN+GFS outperforms the standard GCN, demonstrating the robustness of the TFI approach. For datasets like Fitness and Amazon, there is a noticeable improvement in performance as the supervision percentage increases, but it stabilizes at around $30\%$. In datasets such as History, Photos, and Mines, the model performance is relatively stable, indicating that a small percentage of supervision is sufficient to achieve optimal performance. The performance on Questions shows more variability with increasing supervision, possibly due to the graph-favored dataset's features. This figure supports the claim that TFI requires minimal supervision to improve GCN+GFS performance, which is particularly useful in semi-supervised learning settings.

### D.4  THE NUMBER OF K-HOP NEIGHBORS IN TFI

We further investigate the influence of the number of neighbor hops in TFI on the performance of GCN+GFS. We set $k$ from $\{1, 2, \ldots, 8\}$ as shown in Eq. (5). As illustrated in Figure 14, increasing the number of neighbor hops does not significantly affect model performance and results in only minor deviations in most datasets. This indicates that 1-hop neighbors are sufficient for TFI to select GNN-favored or GNN-disfavored features.

### D.5  PERFORMANCE ON DATASETS WITH PUBLIC SPLITS

We further conduct the experiments on these datasets with public splits as shown in Table 15. Accuracy results on node classification are reported for Cora, PubMed, and CiteSeer, each having one split. For the other datasets, which have ten splits, we provide both accuracy and standard deviation. The results demonstrate that GNN+GFS outperforms baseline GNN in most datasets for 4 types of GNN backbones, which include GCN, GAT, SGC, and GraphSAGE.

Table 15: The performance of graph feature selection on GNN baselines in public split datasets.

| Model | Cora | Pubmed. | Citeseer | Actor | Chameleon. | Cornell | Squirrel | Texas. | Wisconsin |
|---|---|---|---|---|---|---|---|---|---|
| GCN | 81.28 | 78.06 | 70.60 | 33.68±0.64 | 62.41±1.97 | 60.81±6.53 | 48.41±1.52 | 66.49±6.14 | 77.65±4.91 |
| GCN+GFS | 81.94 | 78.46 | 71.74 | 35.30±0.85 | 62.68±2.02 | 68.11±6.47 | 49.13±1.78 | 77.57±6.50 | 80.39±4.89 |
| $\Delta$ | +0.66 | +0.40 | +1.14 | +1.62 | +0.27 | +7.30 | +0.72 | +11.08 | +2.74 |
| GAT | 80.16 | 76.92 | 69.88 | 33.07±0.79 | 65.79±2.55 | 63.24±4.97 | 51.63±1.47 | 74.05±4.97 | 76.47±3.70 |
| GAT+GFS | 81.06 | 77.94 | 72.24 | 35.23±0.83 | 66.49±2.11 | 69.19±5.87 | 52.15±1.83 | 77.84±4.38 | 81.57±6.21 |
| $\Delta$ | +0.90 | +1.02 | +2.36 | +2.16 | +0.70 | +5.95 | +0.52 | +3.79 | +5.10 |
| SGC | 80.74 | 76.96 | 69.84 | 30.64±0.75 | 54.39±1.79 | 43.24±7.64 | 39.91±1.56 | 54.86±4.42 | 53.73±5.64 |
| SGC+GFS | 81.48 | 76.38 | 71.18 | 35.62±1.20 | 54.63±2.14 | 67.30±4.84 | 42.19±1.12 | 71.62±4.27 | 78.63±6.43 |
| $\Delta$ | +0.74 | -0.58 | +1.34 | +4.98 | +0.24 | +24.06 | +2.28 | +16.76 | +24.90 |
| SAGE | 81.06 | 76.80 | 68.52 | 36.20±0.81 | 58.42±2.53 | 70.00±7.03 | 39.32±2.06 | 74.59±5.73 | 81.18±2.11 |
| SAGE+GFS | 81.88 | 78.40 | 71.46 | 36.47±0.72 | 58.71±1.93 | 71.08±5.70 | 39.28±1.49 | 81.08±5.55 | 83.14±5.08 |
| $\Delta$ | +0.82 | +1.60 | +2.94 | +0.27 | +0.29 | +1.08 | -0.04 | +6.49 | +1.96 |

## D.6 IMPACT OF LABEL HOMOPHILY

To investigate how GSF performs under varying label homophily, we conduct experiments on synthetic datasets using CSBM-H (Luan et al., 2024b; Zheng et al., 2024a) to control homophily levels. Specifically, in the CSBM-H model, for a node $u$ labeled with $y$, its features $\mathbf{X}_u \in \mathbb{R}^M$ are sampled from a class-wise Gaussian distribution, specifically $\mathbf{X}_u \sim \mathcal{N}_{Y_u}(\boldsymbol{\mu}_{Y_u}, \boldsymbol{\Sigma}_{Y_u})$. Each dimension of $\mathbf{X}_u$ is independent of each other. To construct the graph structure $\mathcal{G}$ with a specified homophily degree $h$, the node $u$ has a probability $h$ of connecting to nodes with the same label and a probability of $\frac{1-h}{C-1}$ of connecting to nodes with different label. We randomly generate 10 graphs with different seeds in our experiments to reduce uncertainty. Each graph has 1000 nodes with 10 features and 5 classes. The node degrees are uniformly sampled from $[2, 8]$.

We demonstrate how the performance of GCN+GFS varies with label homophily across the range $[0.1, 0.2, \ldots, 0.9]$ in Table 16 and Figure 15 for node classification tasks. Generally, GCN+GFS outperforms both GCN and MLP across different levels of label homophily, indicating that GFS addresses the limitations of GCN under low homophily and of MLP under high homophily through effective feature selection. Furthermore, the relative increase $\Delta$ (defined as the accuracy gap between GCN+GFS and GCN, divided by the accuracy of GCN) decreases as label homophily increases. This finding is expected, as higher homophily levels typically enhance the performance of all graph-aware models, thereby limiting the potential for improvement through GFS.

Table 16: Impact of label homophily on GFS.

| Label Homophily | 0.1 | 0.2 | 0.3 | 0.4 | 0.5 | 0.6 | 0.7 | 0.8 | 0.9 |
|---|---|---|---|---|---|---|---|---|---|
| MLP | 73.12±4.06 | 71.24±5.81 | 68.44±6.93 | 70.36±5.30 | 66.92±6.36 | 67.52±4.64 | 70.12±5.49 | 69.88±4.77 | 69.32±4.68 |
| GCN | 50.12±3.09 | 50.56±3.28 | 56.00±5.49 | 61.24±5.30 | 65.84±5.11 | 72.32±2.35 | 80.32±3.44 | 87.64±2.23 | 90.64±2.32 |
| GCN-GFS | 73.24±3.85 | 69.96±7.24 | 69.04±6.76 | 71.36±6.17 | 69.68±7.69 | 73.84±4.65 | 82.76±6.65 | 89.36±4.20 | 91.20±6.04 |
| $\Delta$ (relative) | 23.12% | 19.40% | 13.04% | 10.12% | 3.84% | 1.52% | 2.44% | 1.72% | 0.56% |

## D.7 IMPACT OF FEATURE DIMENSION

We investigate the impact of feature dimension size based on the CSBM-H model (Luan et al., 2024b; Zheng et al., 2024a), following the setup described in Section D.6. As shown in Table 17 and Figure 16, GFS demonstrates a pronounced advantage over GCN as the feature dimension increases. This outcome is expected, as more features allow GFS to exhibit higher fault tolerance during feature selection. In real-world scenarios, where the feature dimensions in datasets can reach hundreds or thousands, as indicated in Table 3, GFS is likely to perform well.

Table 17: Impact of size of feature dim on GFS.

| #Feature Dim | 10 | 20 | 30 | 40 | 50 |
|---|---|---|---|---|---|
| MLP | 67.52±4.64 | 77.76±3.53 | 83.40±2.58 | 87.60±1.89 | 91.12±1.88 |
| GCN | 72.32±2.35 | 79.20±2.15 | 81.40±2.37 | 84.64±3.07 | 87.80±2.73 |
| GCN-GFS | 73.84±4.65 | 84.12±3.70 | 87.40±3.68 | 90.84±3.51 | 94.12±3.37 |
| $\Delta$ (relative) | 2.06% | 5.85% | 6.86% | 6.83% | 6.71% |

## D.8 Impact of Feature Sparseness

We investigate the impact of feature sparseness on the CSBM-H model (Luan et al., 2024b; Zheng et al., 2024a), following the setup outlined in Section D.6. To control sparseness in synthetic datasets, we randomly mask a certain percentage of node features. As shown in Table 18 and Figure 17, the results indicate that the advantage of GFS diminishes as feature sparseness increases. Although the relative increase rate of GFS decreases in this context, it still outperforms GCN. This is attributed to the presence of both GNN-favored and GNN-disfavored features, which allows GFS to remain effective in enhancing GNN performance.

Table 18: Impact of sparseness on GFS.

| Sparseness | 0% | 10% | 20% | 30% | 40% | 50% | 60% | 70% | 80% | 90% |
|---|---|---|---|---|---|---|---|---|---|---|
| GCN | 54.16±5.58 | 46.48±3.79 | 48.88±3.43 | 45.68±1.93 | 42.12±2.88 | 39.52±2.86 | 34.88±5.59 | 32.08±2.05 | 29.92±2.75 | 25.12±3.07 |
| GCN-GFS | 67.96±7.72 | 59.84±6.83 | 60.04±5.49 | 54.92±5.47 | 50.64±4.15 | 46.20±5.26 | 39.52±6.72 | 35.68±3.05 | 31.36±5.09 | 27.24±3.65 |
| $\Delta$ (relative) | 25.48% | 28.74% | 22.83% | 20.23% | 20.23% | 16.90% | 13.30% | 11.22% | 4.81% | 8.44% |

## D.9 Impact of Noises in Features

To investigate the impact of feature noise on GFS, we conduct experiments on four real-world datasets: Children, Computer, Roman-Empire, and Tolokers, by introducing noise to the original node features. Specifically, after normalizing the input features, we add Gaussian noise multiple times. As shown in Table 19 and Figure 18, the relative increase $\Delta$ (defined as the accuracy gap between GCN+GFS and GCN, divided by the accuracy of GCN) remains relatively stable, indicating that the performance of GFS is robust against feature noise.

Table 19: Impact of noise on GFS.

| Datasets | Noise | 1 | 5 | 10 | 15 | 20 |
|---|---|---|---|---|---|---|
| Children | GCN | 52.92±1.17 | 52.93±0.62 | 52.57±0.75 | 52.32±0.89 | 52.20±0.75 |
| | GCN-GFS | 58.16±1.82 | 57.56±2.64 | 57.01±3.24 | 56.31±3.78 | 55.87±3.94 |
| | $\Delta$ (relative) | 9.89% | 8.75% | 8.44% | 7.65% | 7.04% |
| Comp. | GCN | 84.39±1.25 | 84.42±0.85 | 83.07±1.09 | 82.46±0.77 | 81.68±0.94 |
| | GCN-GFS | 88.87±2.82 | 88.50±2.25 | 87.80±3.17 | 87.10±3.40 | 85.01±5.18 |
| | $\Delta$ (relative) | 5.31% | 4.84% | 5.69% | 5.63% | 4.08% |
| Roman. | GCN | 84.11±0.30 | 83.96±0.33 | 83.65±0.40 | 83.23±0.33 | 83.00±0.25 |
| | GCN-GFS | 84.85±0.80 | 84.57±0.77 | 84.19±0.92 | 83.73±0.91 | 83.59±0.83 |
| | $\Delta$ (relative) | 0.87% | 0.73% | 0.64% | 0.60% | 0.71% |
| Tolokers | GCN | 82.83±1.41 | 83.28±0.73 | 82.81±0.79 | 82.50±0.77 | 82.12±0.62 |
| | GCN-GFS | 86.52±1.85 | 85.99±1.79 | 85.48±1.54 | 84.99±0.87 | 84.44±1.75 |
| | $\Delta$ (relative) | 4.46% | 3.25% | 3.23% | 3.01% | 2.81% |

## D.10 Good Heterophily on feature homophily

In this section, we show the phenomenon of "good heterophily" occurs in feature homophily. First, we split node features into 10 bins according to the values of feature homophily. Then, we run GCN on these bins separately to see how the model performance changes with feature homophily. As shown in Table 20 and Figure 19, the GCN performance remains good under a low value of feature homophily, which includes attribute homophily (Yang et al., 2021) ($h_{attr}$), local similarity (Chen et al., 2023) ($h_{sim-euc}$), and class-controlled feature homophily (Lee et al., 2024) ($h_{CTF}$). This result indicates these consistency-based feature homophily cannot align well with GNN performance, which is similar to the phenomenon in label homophily (Ma et al., 2021; Luan et al., 2024b; Zheng et al., 2024a). Conversely, GNN performance consistently increases with the increase of our proposed TFI, implying its effectiveness in selecting GNN-favored or GNN-disfavored features.

## D.11 Statistical of Selected Features

As shown in Table 21, we present the statistics of values, sparsity, and TFI for all features, GNN-favored features, and GNN-disfavored features. The results indicate that: (1) the TFI of GNN-

Table 20: GCN performance on different bins of feature homophily metrics.

| Metrics | 0.0-0.1 | 0.1-0.2 | 0.2-0.3 | 0.3-0.4 | 0.4-0.5 | 0.5-0.6 | 0.6-0.7 | 0.7-0.8 | 0.8-0.9 | 0.9-1.0 |
|---|---|---|---|---|---|---|---|---|---|---|
| $h_{attr}$ | 31.94±0.16 | 31.51±0.95 | 30.26±4.82 | 30.21±4.76 | 30.02±4.99 | 30.07±3.95 | 29.34±4.30 | 30.85±2.59 | 29.96±3.90 | 31.40±2.26 |
| $h_{sim-euc}$ | 31.92±0.18 | 29.52±5.15 | 31.13±2.02 | 31.21±1.61 | 30.75±3.65 | 30.16±4.75 | 31.38±1.40 | 31.19±1.86 | 31.22±1.61 | 30.89±3.10 |
| $h_{CTF}$ | 31.94±0.16 | 31.43±0.94 | 30.31±4.46 | 30.55±3.37 | 30.79±2.97 | 31.55±0.92 | 30.08±4.67 | 31.31±1.78 | 30.75±2.75 | 31.92±1.74 |
| $TFI$ | 31.94±0.16 | 35.24±1.02 | 37.43±1.63 | 37.44±1.36 | 41.04±0.70 | 43.22±0.55 | 44.28±0.60 | 45.13±0.30 | 45.12±0.33 | 45.67±0.34 |

favored features is significantly higher than that of GNN-disfavored features; (2) the datasets encoded by Pretrained Language Models (PLMs), including Children, Computers, Fitness, History, and Photo, exhibit similar values in features, yet their TFI varies considerably. Notably, a higher average TFI across all features correlates with better GFS performance; (3) GFS is less effective on datasets with higher sparsity, such as Minesweeper and Tolokers. (4) Datasets with lower homophily tend to identify more features as GNN-disfavored, whereas those with higher homophily identify more features as GNN-favored.

Table 21: Summary Statistics of Values, TFI, and Sparseness for Selected Features.

| Dataset | $X$ | | | $X_G$ | | | $X_{\neg G}$ | | | $r$ |
|---|---|---|---|---|---|---|---|---|---|---|
| | Value | Sparseness | TFI | Value | Sparseness | TFI | Value | Sparseness | TFI | |
| Children | 0.0202±0.4081 | 0.00% | 0.0278±0.0158 | 0.0202±0.4081 | 0.00% | 0.0396±0.0143 | 0.0202±0.4082 | 0.00% | 0.0159±0.0039 | 50% |
| Comp. | 0.0201±0.4068 | 0.00% | 0.0965±0.0232 | 0.0201±0.4063 | 0.00% | 0.1010±0.0239 | 0.0202±0.4090 | 0.00% | 0.0785±0.0021 | 80% |
| Fitness | 0.0195±0.3928 | 0.00% | 0.1841±0.0220 | 0.0195±0.3929 | 0.00% | 0.1906±0.0233 | 0.0195±0.3927 | 0.00% | 0.1688±0.0019 | 70% |
| History | 0.0201±0.4039 | 0.00% | 0.0412±0.0261 | 0.0201±0.4034 | 0.00% | 0.0594±0.0258 | 0.0201±0.4044 | 0.00% | 0.0230±0.0057 | 50% |
| Photo | 0.0202±0.4066 | 0.00% | 0.0680±0.0278 | 0.0201±0.4052 | 0.00% | 0.0817±0.0285 | 0.0202±0.4088 | 0.00% | 0.0475±0.0037 | 60% |
| Amazon. | 0.0003±0.0573 | 0.00% | 0.0914±0.0065 | 0.0003±0.0572 | 0.00% | 0.0957±0.0040 | 0.0003±0.0575 | 0.00% | 0.0850±0.0036 | 60% |
| Mines. | 0.1429±0.3499 | 85.71% | 0.0369±0.0384 | 0.1429±0.3499 | 85.71% | 0.0510±0.0368 | 0.1429±0.3499 | 85.71% | 0.0014±0.0014 | 80% |
| Questions | -0.0007±0.0510 | 15.18% | 0.0074±0.0011 | -0.0007±0.0510 | 15.19% | 0.0077±0.0010 | -0.0007±0.0504 | 15.09% | 0.0055±0.0004 | 90% |
| Roman. | 0.0006±0.0986 | 0.27% | 0.0874±0.0396 | 0.0007±0.1015 | 0.15% | 0.1520±0.0388 | 0.0006±0.0979 | 0.30% | 0.0712±0.0167 | 20% |
| tolokers | 0.3825±0.4567 | 48.03% | 0.0295±0.0139 | 0.3826±0.4565 | 48.06% | 0.0349±0.0135 | 0.3823±0.4570 | 47.98% | 0.0169±0.0005 | 70% |

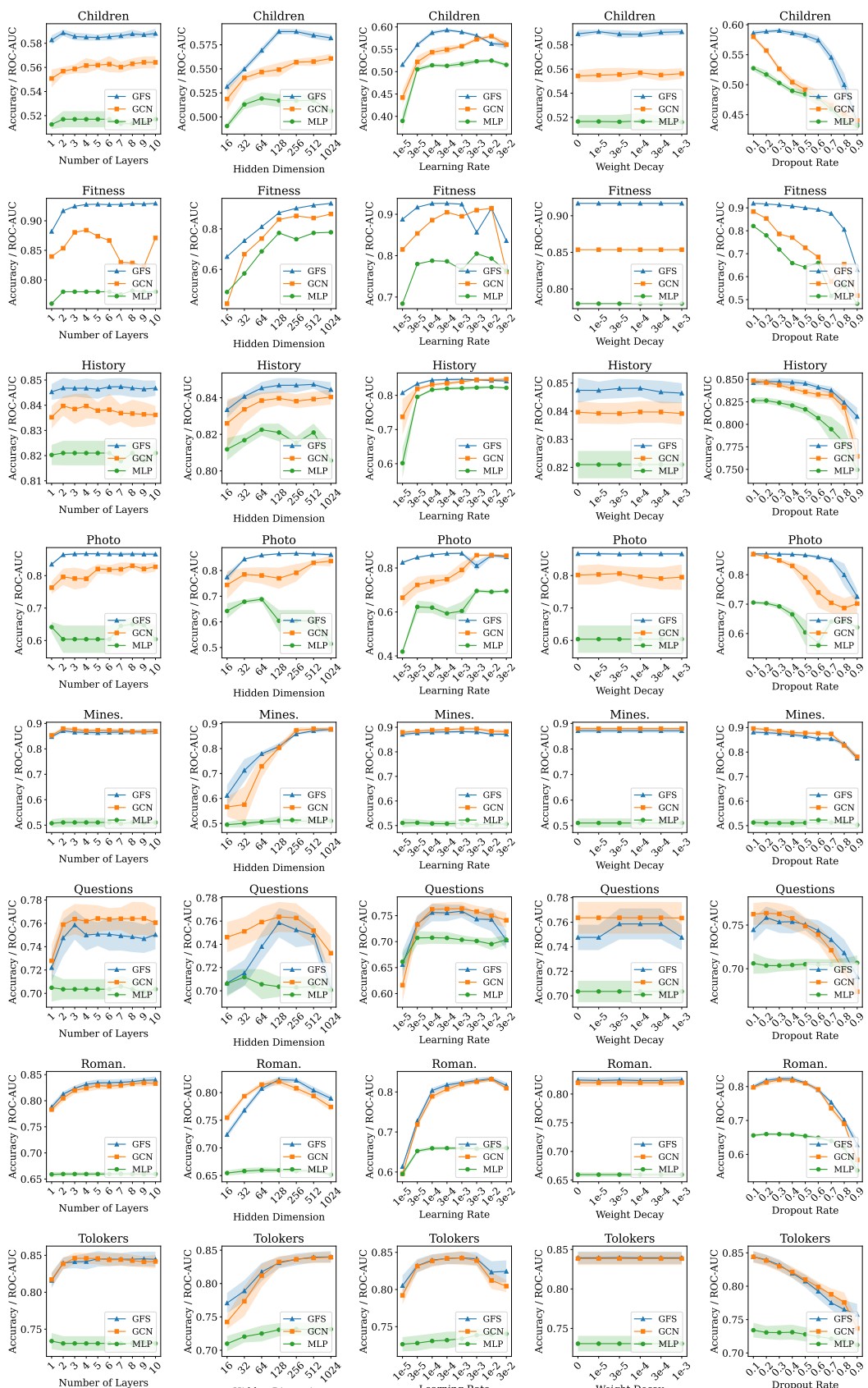

Figure 12: Response of GCN+GFS, GCN, and MLP to 5 hyperparameters on more datasets.

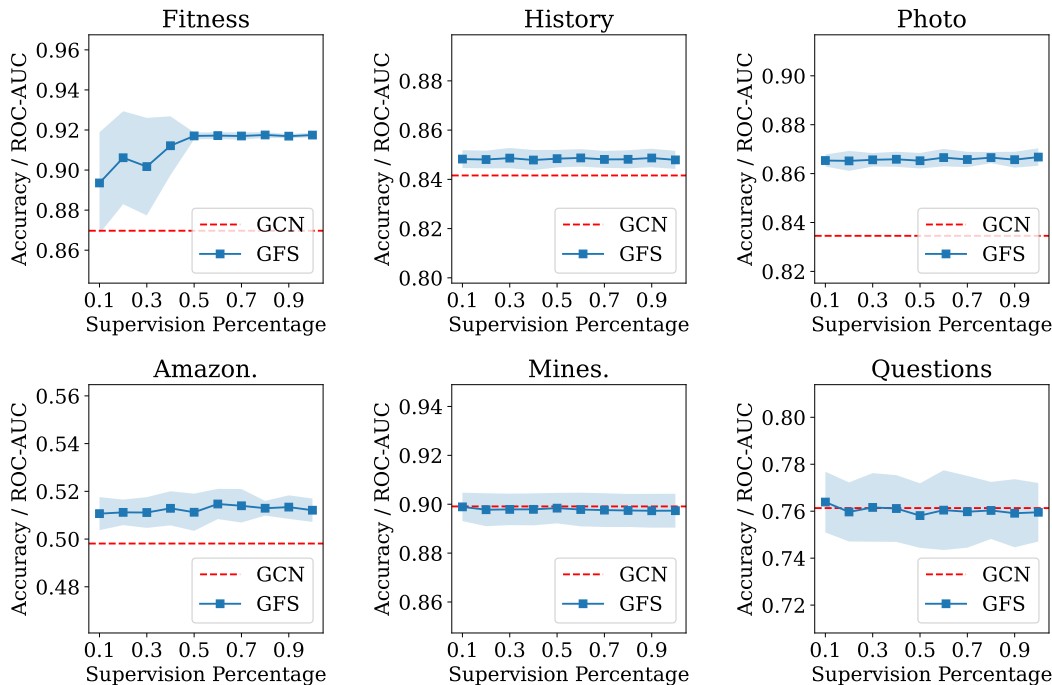

Figure 13: Influence of the percentage of the supervision in TFI on the model performance of GCN+GFS.

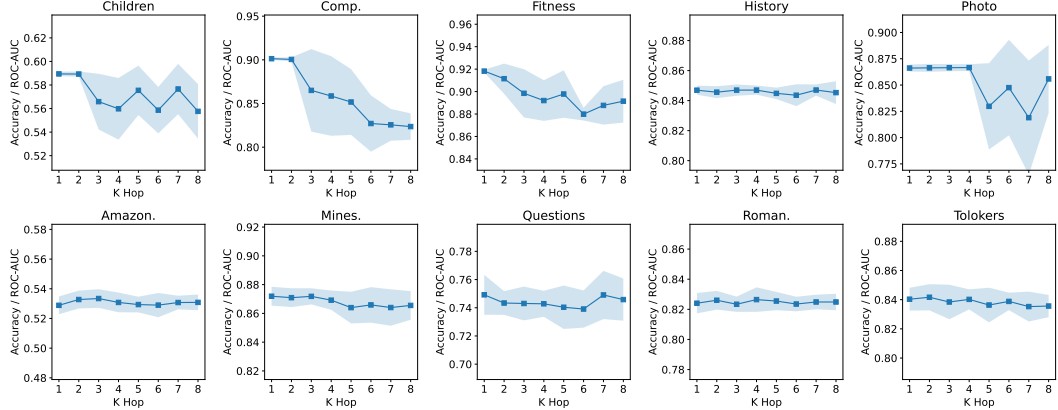

Figure 14: Influence of the number of neighbor hops in TFI on the performance of GCN+GFS.

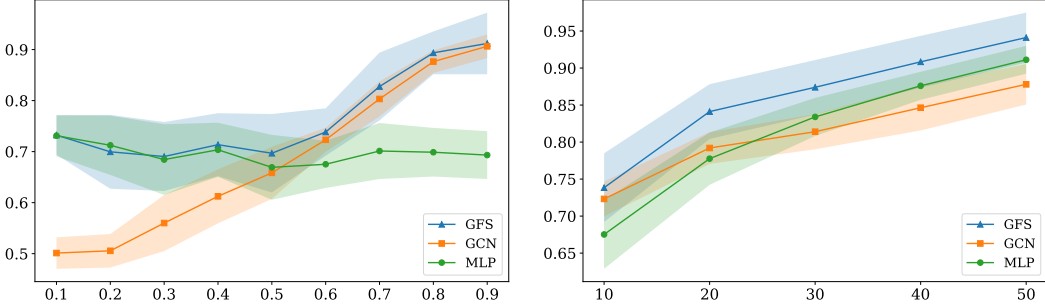

Figure 15: Comparison of the performance of MLP, GCN, and GFS across varying levels of label homophily in synthetic datasets.

Figure 16: Comparison of the performance of MLP, GCN, and GFS with the increase of the number of feature dimensions.

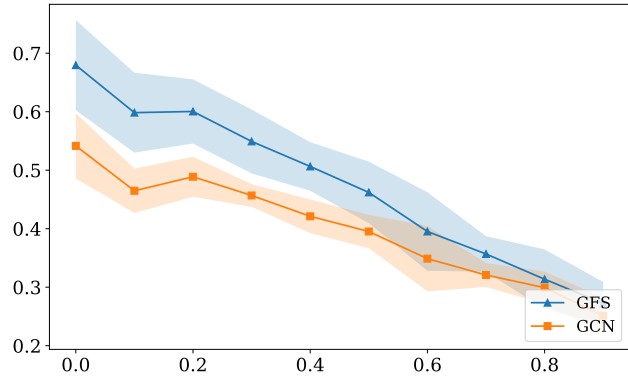

Figure 17: Comparison of the performance of MLP, GCN, and GFS with the sparseness in synthetic datasets

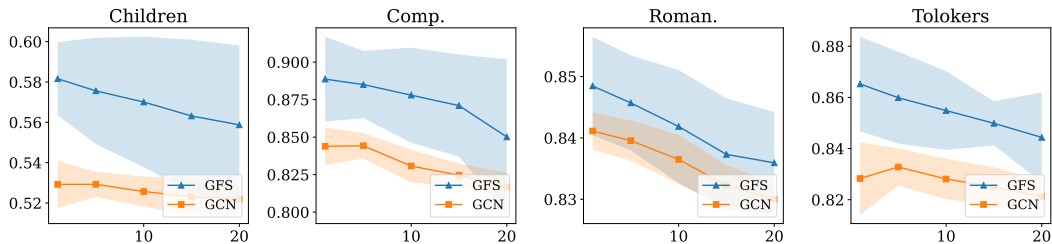

Figure 18: Comparison of the performance of MLP, GCN, and GFS with the increase of feature noises

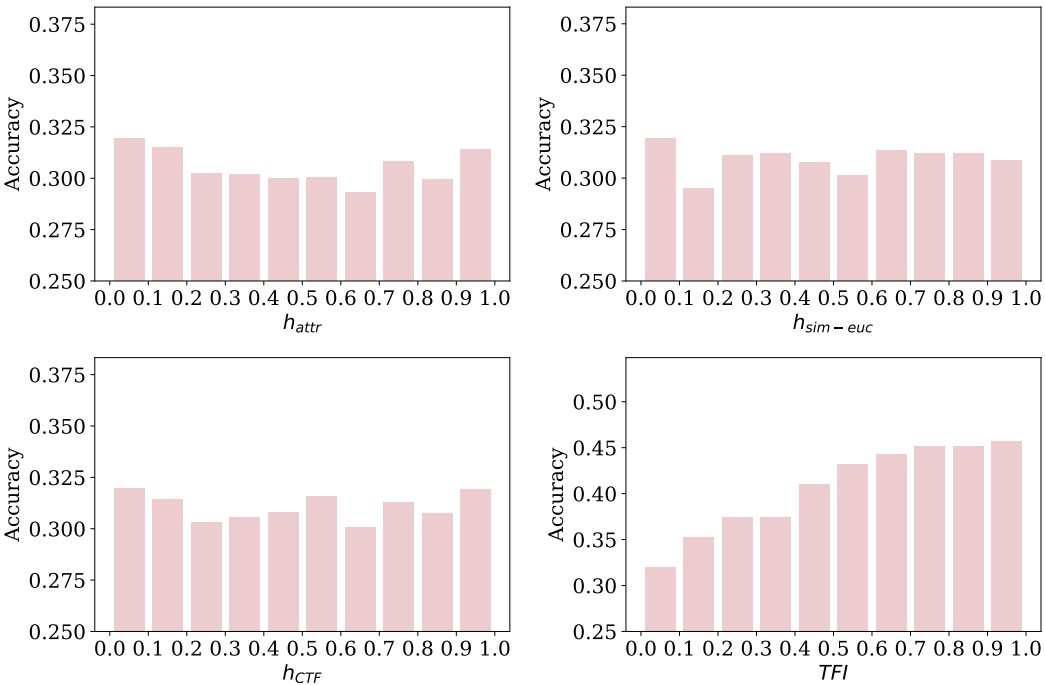

Figure 19: Performance of GCN under varying levels of feature homophily or TFI.

