# OpenReview forum: "Let Your Features Tell The Differences: Understanding Graph Convolution By Feature Splitting"
_ICLR.cc/2025/Conference — ICLR 2025 Poster_

### Official Review · Reviewer_v9Jq · 2024-10-30

**Soundness:** 3
**Presentation:** 3
**Contribution:** 3
**Rating:** 6
**Confidence:** 5

**Summary:**

The paper explores the varying impact of graph convolution across different feature dimensions in GNNs. It introduces a novel metric called Topological Feature Informativeness (TFI), designed to identify features that are either favored or disfavored by GNNs. The authors propose a Graph Feature Selection (GFS) method that leverages TFI to process GNN-favored features with GNNs and GNN-disfavored features with MLPs, enhancing overall performance. The experimental results across multiple GNN architectures and datasets demonstrate that GFS improves accuracy with minimal computational overhead. The work emphasizes the importance of feature-aware processing in GNNs, showing that graph convolution is not uniformly beneficial for all feature dimensions.

**Strengths:**

S1: The concept of distinguishing between GNN-favored and GNN-disfavored features is interesting and novel.

S2: The paper is well-written and structured, easy for readers to follow the methodology and findings.

S3:  The experimental results consistently demonstrate the effectiveness of the proposed GFS trick.

**Weaknesses:**

W1: As far as I am concerned, the theoretical analysis of the proposed strategy is incomplete. The paper does not clearly explain the specific conditions under which distinguishing between GNN-favored and GNN-disfavored features is most effective.

W2: The proposed TFI metric, while beneficial, appears similar to existing mutual information-based methods. The authors need to provide more clarification on how TFI uniquely contributes to feature selection in graph learning, emphasizing its distinct advantages over prior approaches.

W3: Although the experimental results demonstrate strong performance of the GFS-augmented models, I remain somewhat unconvinced. To strengthen the validity of the findings, it would be helpful to include experiments on synthetic datasets, such as those generated by the SBM model with varying levels of heterophily and homophily. Such experiments could offer more reliable insights into the potential advantages of the proposed methodology.

**Questions:**

Q1: I am still unclear about how TFI fundamentally differs from other mutual information-based metrics for feature selection in graph learning. Could the authors provide a clearer explanation of its unique contributions and advantages?

Q2: How does the proposed method handle extremely sparse features, which are common in real-world graphs? Are there specific challenges or limitations in this context?

Q3: It appears that the proposed method offers limited advantages on homophilous graphs compared to its performance on heterophilous graphs. Could the authors provide insights to clarify this discrepancy and explain why the method may be less effective in homophilous settings?

Q4: I am curious about the robustness of the proposed method when applied to noisy graphs, where certain features may contain varying levels of noise. Could the authors provide insights or results on its performance in these scenarios?"

---

> ### Author Response · Authors · 2024-11-21
> **Part (1/4)**
>
> ### W1: As far as I am concerned, the theoretical analysis of the proposed strategy is incomplete. The paper does not clearly explain the specific conditions under which distinguishing between GNN-favored and GNN-disfavored features is most effective.
>
> Thank you for your valuable comments. We do not specify those conditions because features that are either GNN-favored or GNN-disfavored are commonly found in any graph. Only in trivial cases, where all features in a graph are either GNN-favored or GNN-disfavored, is feature selection unnecessary. In all other scenarios, it is essential to perform Graph Feature Selection (GFS) to enhance GNN performance, as demonstrated in our experiments.
>
> Our theoretical analysis demonstrates that TFI acts as an upper bound on the performance gap between graph-aware and graph-agnostic models, establishing the foundation for GFS. While it would be advantageous to theoretically quantify the effectiveness of GFS under specific conditions to facilitate the automatic selection of the optimal threshold rrr, the complexity of measuring the interplay of multiple features during the optimization process of GNNs and MLPs poses significant challenges. We acknowledge this limitation in Section 6. Nonetheless, GFS has already shown substantial improvements across most datasets, and we believe it would be valuable to explore these conditions further in future research.
>
> ### W2: The proposed TFI metric, while beneficial, appears similar to existing mutual information-based methods. The authors need to provide more clarification on how TFI uniquely contributes to feature selection in graph learning, emphasizing its distinct advantages over prior approaches.
>
> As discussed in Appendix A, we highlight the unique contribution of TFI in comparison to other metrics:
>
> 1. Most existing feature selection metrics [3] are designed for non-graph data, making them unsuitable for evaluating the informativeness of features within graphs.
> 2. Traditional methods for feature assessment in graphs rely on consistency-based metrics, which suffer from "good heterophily" issues [1,2] and do not align well with GNN performance.
> 3. TFI stands out as the first method to identify GNN-favored and GNN-disfavored features, effectively reflecting the performance gap between graph-aware and graph-agnostic models, as demonstrated by both theoretical analysis and empirical studies.
>
> We have revised our manuscript to highlight the superiority of TFI over other traditional metrics. Thanks for you suggestions.

---

> ### Author Response · Authors · 2024-11-21
> **Part (2/4)**
>
> ### W3: 1)Although the experimental results demonstrate strong performance of the GFS-augmented models, I remain somewhat unconvinced. 2)To strengthen the validity of the findings, it would be helpful to include experiments on synthetic datasets, such as those generated by the SBM model with varying levels of heterophily and homophily. Such experiments could offer more reliable insights into the potential advantages of the proposed methodology.
>
> First, our proposed GFS is simple yet effective in boosting GNN performance without the need for hyperparameter tuning on most datasets, particularly when features are encoded by Pretrained Language Models (PLMs). This simple method can be easily incorporated into various GNN-related models to enhance performance. Welcome to run our code, which available in the supplementary material or at the anonymous link https://anonymous.4open.science/r/graph-feature-selection-BF28, to verify the effectiveness of GFS.
>
> Second, to further investigate the impact of homophily on GFS, we conducted experiments on synthetic datasets using CSBM-H [1,2] to control homophily levels. Specifically, in the CSBM-H model, for a node $ u $ labeled with $ y $, its features are sampled from a class-wise Gaussian distribution. Each dimension of $ \mathbf{X}_u $ is independent of one another. To construct the graph structure $ \mathcal{G} $ with a specified homophily degree $ h $, node $ u $ has a probability $ h $ of connecting to nodes with the same label and a probability of $ \frac{1-h}{C-1} $ of connecting to nodes with a different label. We randomly generated 10 graphs with different seeds to reduce uncertainty. Each graph contains 1,000 nodes, 10 features, and 5 classes, with node degrees uniformly sampled from the range $[2, 8]$.
>
> | Label Homophily  | 0.1          | 0.2          | 0.3          | 0.4          | 0.5          | 0.6          | 0.7          | 0.8          | 0.9          |
> | ---------------- | ------------ | ------------ | ------------ | ------------ | ------------ | ------------ | ------------ | ------------ | ------------ |
> | **MLP**          | 73.12 ± 4.06 | 71.24 ± 5.81 | 68.44 ± 6.93 | 70.36 ± 5.30 | 66.92 ± 6.36 | 67.52 ± 4.64 | 70.12 ± 5.49 | 69.88 ± 4.77 | 69.32 ± 4.68 |
> | **GCN**          | 50.12 ± 3.09 | 50.56 ± 3.28 | 56.00 ± 5.49 | 61.24 ± 5.30 | 65.84 ± 5.11 | 72.32 ± 2.35 | 80.32 ± 3.44 | 87.64 ± 2.23 | 90.64 ± 2.32 |
> | **GCN+GFS**      | 73.24 ± 3.85 | 69.96 ± 7.24 | 69.04 ± 6.76 | 71.36 ± 6.17 | 69.68 ± 7.69 | 73.84 ± 4.65 | 82.76 ± 6.65 | 89.36 ± 4.20 | 91.20 ± 6.04 |
> | **Δ (relative)** | 23.12%       | 19.40%       | 13.04%       | 10.12%       | 3.84%        | 1.52%        | 2.44%        | 1.72%        | 0.56%        |
>
> We demonstrate how the performance of GCN+GFS varies with label homophily across the range $[0.1, 0.2, \ldots, 0.9]$ in Table 16 and Figure 14 for node classification tasks. Generally, GCN+GFS outperforms both GCN and MLP across various levels of label homophily, indicating that GFS effectively addresses the limitations of GCN under low homophily and of MLP under high homophily through effective feature selection. Furthermore, the relative increase $ \Delta $ (defined as the accuracy gap between GCN+GFS and GCN, divided by the accuracy of GCN) decreases as label homophily increases. This finding is expected, as higher homophily levels typically enhance the performance of all graph-aware models, thereby limiting the potential for improvement through GFS. Furthermore, this conclusion is also verified by our experiments on real-world datasets, including both the homophilous and heterophilous graphs, as shown in Table 1.
>
> Thank you for your insightful suggestions on synthetic datasets.

---

> ### Author Response · Authors · 2024-11-21
> **Part (3/4)**
>
> ### Q1: I am still unclear about how TFI fundamentally differs from other mutual information-based metrics for feature selection in graph learning. Could the authors provide a clearer explanation of its unique contributions and advantages?
>
> In conclusion, previous studies on mutual information (MI) have primarily focused on selecting informative features or reducing redundancy for i.i.d. data in graph-agnostic models (MLP). In contrast, our approach emphasizes distinguishing between features favored by graph-agnostic models (MLP) and those favored by graph-aware models (GNN), which is essential for effective graph representation learning. For more details, please refer to our detailed response to W2.
>
> ### Q2: How does the proposed method handle extremely sparse features, which are common in real-world graphs? Are there specific challenges or limitations in this context?
>
> Thank you for your constructive comments. We further conduct experiments to investigate the impact of feature sparseness on the CSBM-H model [1,2], following the setup outlined in Section D.6 (or responses to W3). To control sparseness in synthetic datasets, we randomly mask a certain percentage of node features. As shown in Table 18 and Figure 16, the results indicate that the advantage of GFS diminishes as feature sparseness increases. Although the relative increase rate of GFS decreases in this context, it still outperforms GCN. This is because GNN-favored and GNN-disfavored features still exist under varying levels of sparseness, which allows GFS to remain effective in enhancing GNN performance.
>
> | Sparseness       | 0%          | 10%         | 20%         | 30%         | 40%         | 50%         | 60%         | 70%         | 80%         | 90%         |
> |------------------|--------------|--------------|--------------|--------------|--------------|--------------|--------------|--------------|--------------|--------------|
> | **GCN**          | 54.16 ± 5.58 | 46.48 ± 3.79 | 48.88 ± 3.43 | 45.68 ± 1.93 | 42.12 ± 2.88 | 39.52 ± 2.86 | 34.88 ± 5.59 | 32.08 ± 2.05 | 29.92 ± 2.75 | 25.12 ± 3.07 |
> | **GCN+GFS**     | 67.96 ± 7.72 | 59.84 ± 6.83 | 60.04 ± 5.49 | 54.92 ± 5.47 | 50.64 ± 4.15 | 46.20 ± 5.26 | 39.52 ± 6.72 | 35.68 ± 3.05 | 31.36 ± 5.09 | 27.24 ± 3.65 |
> | **Δ (relative)** | 25.48%       | 28.74%       | 22.83%       | 20.23%       | 20.23%       | 16.90%       | 13.30%       | 11.22%       | 4.81%        | 8.44%        |
>
> ### Q3: It appears that the proposed method offers limited advantages on homophilous graphs compared to its performance on heterophilous graphs. Could the authors provide insights to clarify this discrepancy and explain why the method may be less effective in homophilous settings?
>
> We agree with your opinion. From both the experiments from real-world datasets (Table 1) and synthetic datasets (Figure 14 and Table 16), we observe that GFS has stronger superior on baseline GNNs. We think this phenomenon mainly comes from two reasons:
>
> 1. Limited potential of increase. The GNN performance becomes better with the increase of label homophily, lefting limited space for improving model performance. A graph with high label homophily indicates that all the nodes group into several class clusters, making it easier to correctly to classify nodes in graph. Give such an easy task, it is not surprising that GFS derives limited superiority.
> 2. Most features are GNN-favored for some datasets. As shown in Figure 4, for homophilous graphs, such as Tolokers, most features are GNN-favored, making GFS have limited increase on model performance. However, this case is very few in all the datasets.
>
> Even if the improvement of GFS, as shown in Figure 4, on heterophilous graphs (Computers and Tolokers) is not as significant as homophilous graphs (Children and Roman-empire), it **still improves the performance of baseline GNN with a higher ratio of $r$ since there still exist some features that are GNN-disfavored**. This indicates the necessity of applying GFS in graphs, which is also verified by additional results on synthetic datasets in Figure 14 and Table 16.

---

> ### Author Response · Authors · 2024-11-21
> **Part (4/4)**
>
> ### Q4: I am curious about the robustness of the proposed method when applied to noisy graphs, where certain features may contain varying levels of noise. Could the authors provide insights or results on its performance in these scenarios?"
>
> We conduct additional experiments to investigate the impact of feature noise on GFS on four real-world datasets: Children, Computer, Roman-Empire, and Tolokers, by introducing noises to the normalized original node features. Specifically, after normalizing the input features, we add Gaussian noise $\epsilon \sim N(0,1)$ for multiple times. As shown in Table 19 and Figure 17, the relative increase $\Delta$ (defined as the accuracy gap between GCN+GFS and GCN, divided by the accuracy of GCN) remains relatively stable, indicating that the performance of GFS is robust against feature noise. We also observe that the superiority of GFS gradually diminishes as feature noise levels increase. We speculate that this is due to the added noise reducing the informativeness of all features, leaving effective information primarily derived from graph topology. As a result, all features may become unsuitable for graph-agnostic models, collapsing into features favored by GNNs, which diminishes the effectiveness of GFS. However, it is important to note that adding noise to all features more than 20 times is uncommon in real-world scenarios. Therefore, GFS remains effective on the majority of datasets, as demonstrated in Table 1. Thank you very much for your valuable suggestions.
>
> | Datasets   | Noise |     1       |     5       |     10      |     15      |     20      |
> |------------|-------|-------------|-------------|-------------|-------------|-------------|
> | **Children** | GCN   | 52.92±1.17  | 52.93±0.62  | 52.57±0.75  | 52.32±0.89  | 52.20±0.75  |
> |            | GCN+GFS | 58.16±1.82 | 57.56±2.64  | 57.01±3.24  | 56.31±3.78  | 55.87±3.94  |
> |            | Δ (relative) | 9.89%   | 8.75%       | 8.44%       | 7.65%       | 7.04%      |
> | **Comp.**  | GCN   | 84.39±1.25  | 84.42±0.85  | 83.07±1.09  | 82.46±0.77  | 81.68±0.94  |
> |            | GCN+GFS | 88.87±2.82 | 88.50±2.25  | 87.80±3.17  | 87.10±3.40  | 85.01±5.18  |
> |            | Δ (relative) | 5.31%   | 4.84%       | 5.69%       | 5.63%       | 4.08%       |
> | **Roman.** | GCN   | 84.11±0.30  | 83.96±0.33  | 83.65±0.40  | 83.23±0.33  | 83.00±0.25  |
> |            | GCN+GFS | 84.85±0.80 | 84.57±0.77  | 84.19±0.92  | 83.73±0.91  | 83.59±0.83  |
> |            | Δ (relative) | 0.87%   | 0.73%       | 0.64%       | 0.60%       | 0.71%       |
> | **Tolokers** | GCN   | 82.83±1.41  | 83.28±0.73  | 82.81±0.79  | 82.50±0.77  | 82.12±0.62  |
> |            | GCN+GFS | 86.52±1.85 | 85.99±1.79  | 85.48±1.54  | 84.99±0.87  | 84.44±1.75  |
> |            | Δ (relative) | 4.46%   | 3.25%       | 3.23%       | 3.01%       | 2.81%       |
>
> [1] Ma Y, Liu X, Shah N, et al. Is homophily a necessity for graph neural networks?[J]. arXiv preprint arXiv:2106.06134, 2021.
>
> [2] Luan S, Hua C, Xu M, et al. When do graph neural networks help with node classification? investigating
>
> [3] Feature Selection: A Data Perspective. ACM Comput. Surv. 50(6).

---

> ### Author Response · Authors · 2024-11-28
>
> Please let me know if our responses have addressed your concerns. Thank you!

---

### Official Review · Reviewer_poRT · 2024-10-31

**Soundness:** 3
**Presentation:** 3
**Contribution:** 3
**Rating:** 6
**Confidence:** 4

**Summary:**

GNNs have strong graph learning capabilities. However, GNNs are not effective at learning all graph data features. In previous work, many metrics based on graph topology and features have been proposed to describe feature homogeneity in order to assess the effectiveness of GNNs. However, there has been no metric that can directly and effectively guide which features GNNs should learn. To fill this gap, this paper proposes an evaluation metric, Topological Feature Informative (TFI), to compute the learnability of features for GNNs. Subsequently, a dual-channel embedding architecture combining GNNs and MLPs is used to embed these features separately.

**Strengths:**

1. The writing of the paper is clear and easy to understand, with a well-defined theme.
2. The paper provides a theoretical foundation for using TFI to guide feature selection through simple proofs.
3. The experimental results on node classification tasks are impressively good.

**Weaknesses:**

The paper does not provide any statistical analysis of the selected features, which raises some questions.

**Questions:**

1. How is the decomposition of node features in the graph executed?
2. What impact does the dimension of the initial node features have?
3. Can this method only be applied to node features, or can it also be used on edge features, and what would be the effect?
4. Is it possible to measure what a "good heterophily" feature looks like? Can other metrics be used for supplementary description, such as Label Homophily, Feature Homophily, Mutual Information, and so on?

---

> ### Author Response · Authors · 2024-11-21
> **Part (1/3)**
>
> ### W1: The paper does not provide any statistical analysis of the selected features, which raises some questions.
>
> Thank you for your valuable feedback. We agree that a more detailed statistical analysis of these features would be beneficial. In our submission, we provide an analysis of the selected features:
>
> - In Figures 4 and 10, we highlight the ratio of selected features in GFS.
> - Figure 8 demonstrates that swapping GNN-favored and GNN-disfavored features leads to a decrease in GFS performance.
>
> Additional, as shown in Table 21, we present the statistics of values, sparsity, and TFI for all features, GNN-favored features, and GNN-disfavored features. The results indicate that: (1) the TFI of GNN-favored features is significantly higher than that of GNN-disfavored features; (2) the datasets encoded by Pretrained Language Models (PLMs), including Children, Computers, Fitness, History, and Photo, exhibit similar values in features, yet their TFI varies considerably. Notably, a higher average TFI across all features correlates with better GFS performance; (3) GFS is less effective on datasets with higher sparsity, such as Minesweeper and Tolokers. (4) Datasets with lower homophily tend to identify more features as GNN-disfavored, whereas those with higher homophily identify more features as GNN-favored.
>
> | Dataset   | Value            | Sparseness | TFI             | Value            | Sparseness | TFI             | Value            | Sparseness | TFI             | r    |
> | --------- | ---------------- | ---------- | --------------- | ---------------- | ---------- | --------------- | ---------------- | ---------- | --------------- | ---- |
> | Children  | 0.0202 ± 0.4081  | 0.00%      | 0.0278 ± 0.0158 | 0.0202 ± 0.4081  | 0.00%      | 0.0396 ± 0.0143 | 0.0202 ± 0.4082  | 0.00%      | 0.0159 ± 0.0039 | 50%  |
> | Comp.     | 0.0201 ± 0.4068  | 0.00%      | 0.0965 ± 0.0232 | 0.0201 ± 0.4063  | 0.00%      | 0.1010 ± 0.0239 | 0.0202 ± 0.4090  | 0.00%      | 0.0785 ± 0.0021 | 80%  |
> | Fitness   | 0.0195 ± 0.3928  | 0.00%      | 0.1841 ± 0.0220 | 0.0195 ± 0.3929  | 0.00%      | 0.1906 ± 0.0233 | 0.0195 ± 0.3927  | 0.00%      | 0.1688 ± 0.0019 | 70%  |
> | History   | 0.0201 ± 0.4039  | 0.00%      | 0.0412 ± 0.0261 | 0.0201 ± 0.4034  | 0.00%      | 0.0594 ± 0.0258 | 0.0201 ± 0.4044  | 0.00%      | 0.0230 ± 0.0057 | 50%  |
> | Photo     | 0.0202 ± 0.4066  | 0.00%      | 0.0680 ± 0.0278 | 0.0201 ± 0.4052  | 0.00%      | 0.0817 ± 0.0285 | 0.0202 ± 0.4088  | 0.00%      | 0.0475 ± 0.0037 | 60%  |
> | Amazon.   | 0.0003 ± 0.0573  | 0.00%      | 0.0914 ± 0.0065 | 0.0003 ± 0.0572  | 0.00%      | 0.0957 ± 0.0040 | 0.0003 ± 0.0575  | 0.00%      | 0.0850 ± 0.0036 | 60%  |
> | Mines.    | 0.1429 ± 0.3499  | 85.71%     | 0.0369 ± 0.0384 | 0.1429 ± 0.3499  | 85.71%     | 0.0510 ± 0.0368 | 0.1429 ± 0.3499  | 85.71%     | 0.0014 ± 0.0014 | 80%  |
> | Questions | -0.0007 ± 0.0510 | 15.18%     | 0.0074 ± 0.0011 | -0.0007 ± 0.0510 | 15.19%     | 0.0077 ± 0.0010 | -0.0007 ± 0.0504 | 15.09%     | 0.0055 ± 0.0004 | 90%  |
> | Roman.    | 0.0006 ± 0.0986  | 0.27%      | 0.0874 ± 0.0396 | 0.0007 ± 0.1015  | 0.15%      | 0.1520 ± 0.0388 | 0.0006 ± 0.0979  | 0.30%      | 0.0712 ± 0.0167 | 20%  |
> | Tolokers  | 0.3825 ± 0.4567  | 48.03%     | 0.0295 ± 0.0139 | 0.3826 ± 0.4565  | 48.06%     | 0.0349 ± 0.0135 | 0.3823 ± 0.4570  | 47.98%     | 0.0169 ± 0.0005 | 70%  |
>
> ### Q1: How is the decomposition of node features in the graph executed?
>
> We would like to clarify that in GFS, we do not decompose node features. Instead, we categorize the node features into two sets: GNN-favored and GNN-disfavored features, based on TFI. This categorization is guided by the principle of "feeding the right features to the right model." To assess how GNN or MLP benefits from specific features, we utilize our proposed TFI and apply a threshold to classify all features into GNN-favored and GNN-disfavored categories. The effectiveness of TFI has been validated through both theoretical analysis and empirical studies. Thank you for your comments. We have revised the manuscript to enhance clarity and improve understanding.

---

> ### Author Response · Authors · 2024-11-21
> **Part (2/3)**
>
> ### Q2: What impact does the dimension of the initial node features have?
>
> Sorry, we are unclear about your question. Are you referring to the impact of the number of feature dimensions? This is a good question. The number of feature dimensions does influence the performance of GFS. In Table 1, we observe that datasets with a higher number of features, such as Children (dim=768), Computer (dim=768), and Fitness (dim=768), show greater improvements in GFS performance compared to datasets with fewer features, like Question (dim=301), Roman-empire (dim=300), and Tolokers (dim=10).
>
> Additionally, we conducted experiments to further investigate the impact of feature dimension size on synthetic datasets. As shown in Table 17 and Figure 15, GFS demonstrates a pronounced advantage over GCN as the feature dimension increases. We speculate that this is due to the higher error tolerance of GFS when more features are available, even if some features are misclassified as GNN-favored or GNN-disfavored.
>
> | #Feature Dim | 10           | 20           | 30           | 40           | 50           |
> | ------------ | ------------ | ------------ | ------------ | ------------ | ------------ |
> | MLP          | 67.52 ± 4.64 | 77.76 ± 3.53 | 83.40 ± 2.58 | 87.60 ± 1.89 | 91.12 ± 1.88 |
> | GCN          | 72.32 ± 2.35 | 79.20 ± 2.15 | 81.40 ± 2.37 | 84.64 ± 3.07 | 87.80 ± 2.73 |
> | GCN+GFS      | 73.84 ± 4.65 | 84.12 ± 3.70 | 87.40 ± 3.68 | 90.84 ± 3.51 | 94.12 ± 3.37 |
> | Δ (relative) | 2.06%        | 5.85%        | 6.86%        | 6.83%        | 6.71%        |
>
> It is important to note that most datasets contain hundreds of features, which enhances GFS's robustness and enables it to outperform baseline GNN models, as demonstrated in Table 1. Furthermore, for datasets with fewer features, GFS performance can still be improved by leveraging pretrained node embeddings to increase the feature count. This is illustrated in Figure 7, where GFS performance on Question, Roman-empire, and Tolokers shows significant enhancement. Thank you for your valuable insights. We have revised the paper accordingly.
>
> ### Q3: Can this method only be applied to node features, or can it also be used on edge features, and what would be the effect?
>
> Thank you for your suggestions. We did not apply GFS to edge features for the following reasons:
>
> 1. Lack of Datasets: To the best of our knowledge, there are no graph datasets that incorporate edge features for node classification, as demonstrated in most widely used GNN libraries [1, 2].
> 2. Limited GNN Designs: Consequently, there are not many GNNs designed to handle this type of graph.
>
> For the scenario you mentioned, we believe knowledge graphs are more suitable, as edge features are commonly found in them. In the future, it would be interesting to extend the aggregated node features from $\frac{\sum_{j\in\mathcal{N}_i} X_j}{|\mathcal{N}_i|}$ to $\frac{\sum_{j\in\mathcal{N}_i} f(X_j+r_{ij},X_i)}{|\mathcal{N}_i|}$ [5] to incorporate edge attributes. However, it is important to note that knowledge graphs differ significantly from the pure graphs we studied in this paper. Our main focus is to enhance representation learning on graphs, where performance is typically evaluated through node classification [3, 4]. We follow this setting in our experiments.

---

> ### Author Response · Authors · 2024-11-21
> **Part (3/3)**
>
> ### Q4: Is it possible to measure what a "good heterophily" feature looks like? Can other metrics be used for supplementary description, such as Label Homophily, Feature Homophily, Mutual Information, and so on?
>
> "Good heterophily" [6,7] refers to the phenomenon that traditional consistency-based homophily metrics, such as attribute homophily [9], that cannot align well the GNN performance, reflecting in a good GNN performance under a low homophily value. We further conduct experiments to explain the "good heterophily". First, we split node features into $10$ bins according to the values of feature homophily. Then, we run GCN on these bins separately to see how the model performance changes with feature homophily. As shown in Table 20 and Figure 18, the GCN performance remains good under a low value of feature homophily, which includes attribute homophily [9] ($h_{attr}$), local similarity [10] ($h_{sim-euc}$), and class-controlled feature homophily [8] ($h_{CTF}$). This result indicates these consistency-based feature homophily cannot align well with GNN performance, which is similar to the phenomenon in label homophily [6,7]. Conversely, GNN performance consistently increases with the increase of our proposed TFI, implying its effectiveness in selecting GNN-favored or GNN-disfavored features. Thank you once again for your suggestions. We have revised our paper accordingly.
>
> | Metrics       | 0.0-0.1     | 0.1-0.2     | 0.2-0.3     | 0.3-0.4     | 0.4-0.5     | 0.5-0.6     | 0.6-0.7     | 0.7-0.8     | 0.8-0.9     | 0.9-1.0     |
> |---------------|--------------|--------------|--------------|--------------|--------------|--------------|--------------|--------------|--------------|--------------|
> | $h_{attr}$  | 31.94 ± 0.16 | 31.51 ± 0.95 | 30.26 ± 4.82 | 30.21 ± 4.76 | 30.02 ± 4.99 | 30.07 ± 3.95 | 29.34 ± 4.30 | 30.85 ± 2.59 | 29.96 ± 3.90 | 31.40 ± 2.26 |
> | $h_{sim-euc}$ | 31.92 ± 0.18 | 29.52 ± 5.15 | 31.13 ± 2.02 | 31.21 ± 1.61 | 30.75 ± 3.65 | 30.16 ± 4.75 | 31.38 ± 1.40 | 31.19 ± 1.86 | 31.22 ± 1.61 | 30.89 ± 3.10 |
> | $h_{CTF}$   | 31.94 ± 0.16 | 31.43 ± 0.94 | 30.31 ± 4.46 | 30.55 ± 3.37 | 30.79 ± 2.97 | 31.55 ± 0.92 | 30.08 ± 4.67 | 31.31 ± 1.78 | 30.75 ± 2.75 | 31.92 ± 1.74 |
> | $TFI$       | 31.94 ± 0.16 | 35.24 ± 1.02 | 37.43 ± 1.63 | 37.44 ± 1.36 | 41.04 ± 0.70 | 43.22 ± 0.55 | 44.28 ± 0.60 | 45.13 ± 0.30 | 45.12 ± 0.33 | 45.67 ± 0.34 |
>
>
> [1] Torch Geometric Datasets - pytorch\_geometric documentation.
>
> [2] DGL Datasets - DGL 2.2.1 documentation.
>
> [3] Grover A, Leskovec J. node2vec: Scalable feature learning for networks[C]//Proceedings of the 22nd ACM SIGKDD international conference on Knowledge discovery and data mining. 2016: 855-864.
>
> [4] Chen F, Wang Y C, Wang B, et al. Graph representation learning: a survey[J]. APSIPA Transactions on Signal and Information Processing, 2020, 9: e15.
>
> [5] Bordes A, Usunier N, Garcia-Duran A, et al. Translating embeddings for modeling multi-relational data[J]. Advances in neural information processing systems, 2013, 26.
>
> [6] Ma Y, Liu X, Shah N, et al. Is homophily a necessity for graph neural networks?[J]. arXiv preprint arXiv:2106.06134, 2021.
>
> [7] Luan S, Hua C, Xu M, et al. When do graph neural networks help with node classification? investigating the homophily principle on node distinguishability[J]. Advances in Neural Information Processing Systems, 2024, 36.
>
> [8] Soo Yong Lee, Sunwoo Kim, Fanchen Bu, Jaemin Yoo, Jiliang Tang, and Kijung Shin. Feature
> distribution on graph topology mediates the effect of graph convolution: Homophily perspective.
> International Conference on Machine Learning, 2024
>
> [9] Liang Yang, Mengzhe Li, Liyang Liu, Chuan Wang, Xiaochun Cao, Yuanfang Guo, et al. Diverse message passing for attribute with heterophily. Advances in Neural Information Processing Systems, 34:4751–4763, 2021b.
>
> [10] Yuhan Chen, Yihong Luo, Jing Tang, Liang Yang, Siya Qiu, Chuan Wang, and Xiaochun Cao.
> Lsgnn: towards general graph neural network in node classification by local similarity. arXiv
> preprint arXiv:2305.04225, 2023

---

> ### Author Response · Authors · 2024-11-28
>
> Please let me know if our responses have addressed your concerns. Thank you!

---

### Official Review · Reviewer_iSuK · 2024-11-02

**Soundness:** 2
**Presentation:** 3
**Contribution:** 3
**Rating:** 6
**Confidence:** 3

**Summary:**

The paper introduces a new method to identify features that are favored and disfavored by GNNs. It uses topological feature selection to integrate these features into GNNs, leading to significant improvements in their performance.

**Strengths:**

1. The paper is well-motivated and presents a novel approach to distinguish between GNN-favored and GNN-disfavored features, treating them separately to learn with different methods.
2. The paper is clearly presented and includes solid theoretical guarantees, making it easy to follow.
3. The authors provide sufficient empirical analysis, including ablation studies and comparisons with state-of-the-art methods.

**Weaknesses:**

The evaluation method is not fair enough. The performance results in Table 1 should be presented with only the difference of with or without GPS, while keeping the model architecture the same (e.g., number of layers and hidden dimension).

**Questions:**

1. As shown in Figure 2, the performance gap for the Roman dataset between GCN and MLP in regions with low TFI is minimal. How is the improvement over GCN in Figure 4 so significant regardless of $r$?
2. Since features are updated in each GNN/MLP layer, is the TFI computed at each layer to get (dis)favored features, or is it only computed at the beginning? This point does not seem clearly explained in the paper.

---

> ### Author Response · Authors · 2024-11-21
> **Part (1/1)**
>
> ### W1: The evaluation method is not fair enough. The performance results in Table 1 should be presented with only the difference of with or without GPS, while keeping the model architecture the same (e.g., number of layers and hidden dimension).
>
> Thanks for your valuable comments. Here we need to clarify that **Table 1 shows the results of GNN and GNN+GFS under the same hyperparameter tuning**, which has been widely used to fairly compare the performance of GNNs. As for the **setting you mentioned that keeps the same model architectures, we also show the results in Section 5**: 1) Figure 4 shows GCN+GFS consistently outperforms GCN across a wide range of $r$ (0.1-0.9) in most datasets while keeping all the other hyperparameters, e.g. number of layers and hidden dimension, the same. 2) Figure 5 shows how the performance of GCN+GFS and GCN would be affected under varying hyperparameters where the points in each sub-figure denote they are evaluated under the same hyperparameters. We have revised the paper to help better understand the experimental results. Thank you.
>
> ### Q1: As shown in Figure 2, the performance gap for the Roman dataset between GCN and MLP in regions with low TFI is minimal. How is the improvement over GCN in Figure 4 so significant regardless of $r$ ?
>
> As shown in Figure 2, the accuracy gap (GCN-MLP) is close to 0 except for the bin with high TFI (TFI>0.8). This result **is consistent with the result in Figure 4**, where we only send the features with high TFI (r=0.2) to GCN greatly improves the performance compared with MLP (r=0.2). Note that the x-axis in Figure 2 and Figure 4 represent different meanings: x-axis in Figure 2 means we send 10% of the features with a certain range of TFI to GCN and MLP and compare their performance gap, while the x-axis in Figure 4 means we split all the features into two part, where the highest r% features are sent to GNN and the left features are sent to MLP. Sorry for the confusion, we have revised the paper to enhance clarity and improve understanding.
>
> ### Q2: Since features are updated in each GNN/MLP layer, is the TFI computed at each layer to get (dis)favored features, or is it only computed at the beginning? This point does not seem clearly explained in the paper.
>
> We clarity how the feature selection is conducted in the paper:
>
> 1. In Line 81 in Section 1 (Introduction), we mention that we first use TFI to select GNN-favored and GNN-disfavored features and then process these two types of features by GNN and MLP respectively.
> 2. In Figure 3 In Section 4 (Graph Feature Selection), we show that the features are sorted before sending to GNN and MLP at the beginning, instead of sending node representations.
> 3. In line 257-260 in Section 4 (Graph Feature Selection), we show that we select the node features, i.e. ${\mathbf{X_{\mathcal{G}}} | \mathbf{X_{:, m}} \in \mathbf{X_{\mathcal{G}}}, \text{TFI}_m \ge \delta(r)}$, in GFS. Thanks for your comments, we have further revised the paper to improve the readability. It is a good question. As shown in Figure 7 in our submission, we conduct the GFS on node embeddings by different approaches to see if it will also improve the performance of baseline GNN. The results show that in some datasets, such as Questions, Roman-empire, and Tolokers, performing GFS on node embeddings further improves the GFS performance. It is interesting to investigate how the GFS affects the model performance on node embeddings encoded by more advanced models in the future.
>
> We greatly appreciate your valuable reviews. Thank you.

---

> ### Comment · Reviewer_iSuK · 2024-11-27
>
> Thank you for your response. I am willing to improve my score. However, I still have concerns regarding my initial question. From what I understand, dividing the entire graph into 10 parts to create subgraphs and then training MLP or GNN separately on these subgraphs could potentially ignore the global information, which is essential for GNNs. To my understanding, an effective method would align with the approach depicted in Figure 4, where only the nodes ranging from $x$ to $x+0.1$ are processed by the MLP, while the rest of the graph remains unchanged. This would enable a more precise analysis of the performance gap in each bin.

---

> ### Author Response · Authors · 2024-11-27
>
> Thank you for your positive rating. We’d like to clarify that our method does not divide the entire graph into different parts; rather, it categorizes node features into GNN-favored and GNN-disfavored groups while keeping the graph structure unchanged. By splitting all node features into 10 bins, Figure 2 illustrates how TFI can identify features that are favored by both graph-aware and graph-agnostic models. These findings support our experimental results in Figure 4, where GFS demonstrates superiority over baseline GNNs by adhering to the principle of "feeding the right features to the right models". Besides, GFS with r = 0.1 to 0.9 consistently outperforms the models that send all the features to GCN (r = 1.0) or MLP (r = 0.0). We will revise the paper to enhance its clarity and make it easier to understand. Thank you.

---

> > ### Comment · Reviewer_iSuK · 2024-11-27
> >
> > Thank you for your clarification. It makes sense, and I am looking forward to a clearer description of Figure 2.

---

> > > ### Author Response · Authors · 2024-11-28
> > >
> > > We have revised our paper in the submission. Thank you!

---

### Official Review · Reviewer_jpAB · 2024-11-03

**Soundness:** 3
**Presentation:** 3
**Contribution:** 2
**Rating:** 5
**Confidence:** 4

**Summary:**

This paper investigates the problem of feature selection for graph Convolution Networks (GCNs). It begins by introducing TFI, a metric designed to guide the selection of relevant features. Following that, it introduces GFS, a plug-in method that distinguishes between features that benefit graph convolution and those that do not contribute positively or may even have a negative impact. Then, the two sets of features are processed separately using GCN and Multi-Layer Perceptrons (MLP), respectively. Evaluations on node classification tasks demonstrate the effectiveness of the proposed TFI and GFS.

**Strengths:**

1 This paper is well-organized and easy to follow.

2 The figures regarding the design motivation and the proposed framework (Figures 1 and 3) are clear.

3 The theoretical proofs are detailed.

**Weaknesses:**

1 The proposed feature selection metric TFI, which leverages mutual information between features and labels, is not novel. This is a conventional approach for feature selection, as outlined in [1]. Although the TFI utilizes the features derived from the neighborhood average (AX), the approach is incremental.

2 This paper lacks a comparative analysis of classic feature selection methods, such as [2].

3 The GPS exhibits limited robustness to the selection of the hyperparameter r.

4 It is unclear how the high-pass filters of FAGCN and ACMGNN, as presented in Table 1, align with the TFI metric with low-pass AX.

5 A significant concern is the applicability of the proposed GFS. The paper asserts that TFI is computed on training nodes. Thus, I am concerned that the unusual dataset division of 50/25/25 for training/validation/testing employed in this paper is crucial for the effectiveness of GFS. The question then becomes: how would the results be affected if a public splitting, such as 20 per class for training in GCN, SGC, GAT, and APPNP, were applied?

[1] Feature Selection: A Data Perspective. ACM Comput. Surv. 50(6).

[2] Multi-label Feature Selection via Global Relevance and Redundancy Optimization. IJCAI 2020.

**Questions:**

See Weaknesses.

---

> ### Author Response · Authors · 2024-11-21
> **Part(1/2)**
>
> ### W1: The proposed feature selection metric TFI, which leverages mutual information between features and labels, is not novel. This is a conventional approach for feature selection, as outlined in [1]. Although the TFI utilizes the features derived from the neighborhood average (AX), the approach is incremental.
>
> Thank you for your review. Below are our responses to your concerns regarding the novelty of our work:
>
> - As detailed in Appendix A, we clarify the novelty of our method in contrast to traditional approaches. Previous studies on mutual information (MI) [1] have primarily focused on selecting the most informative features or reducing redundancy for independent and identically distributed (i.i.d.) data in graph-agnostic models (MLP). In contrast, our approach emphasizes distinguishing between features favored by graph-agnostic models (MLP) and those favored by graph-aware models (GNN), which is essential for effective graph representation learning.
>
> - In addition to introducing the novel metric, TFI, this paper presents a framework called Graph Feature Selection (GFS). This framework effectively leverages the strengths of both GNN and non-GNN models, demonstrating simplicity and effectiveness in enhancing the performance of eight GNNs across ten datasets. To the best of our knowledge, we are the first to investigate feature selection in the context of both GNN and non-GNN models.
>
> - In the context of graphs, TFI quantifies the mutual information between aggregated features and labels. Although this method is straightforward and intuitive, it is effective in identifying GNN-favored and GNN-disfavored features, as evidenced by our theoretical analyses and empirical studies.
>
> - The extension of aggregated features is a non-trivial task, as the field of graph learning is evolving to analyze the transformation from X to AX. This extension clarifies the role of topological information in GNNs and is crucial for understanding the relationships and interactions within graphs.
>
> We have revised our paper to highlight our novelty. Thank you for your suggestions.
>
> ### W2: This paper lacks a comparative analysis of classic feature selection methods, such as [2].
>
> Our primary focus is on investigating feature selection across both graph and non-graph data, rather than proposing a feature selection method specifically for non-graph data. Consequently, we do not compare the performance of classic feature selection methods that are designed for non-graph data. For instance, reference [2] is designed for non-graph data in a multi-label setting, which differs significantly from our context. As shown in Table 2, we do compare metrics such as generalized feature homophily [3], attributed homophily [4], and local similarity [5], which are specifically designed for graph data. We appreciate your suggestions regarding these interesting metrics and believe that incorporating additional metrics into GFS could further enhance graph representation learning.
>
> ### W3: The GPS exhibits limited robustness to the selection of the hyperparameter r.
>
> As shown in Figure 4, the performance of GFS remains robust across varying hyperparameter values of $r$. GFS consistently outperforms the baseline GNN across a wide range of $r$, from 0.1 to 0.9, in most datasets, including Children, Computer, and Roman-Empire.
>
> ### W4: It is unclear how the high-pass filters of FAGCN and ACMGNN, as presented in Table 1, align with the TFI metric with low-pass AX.
>
> It is a good question. In this paper, we treat GNNs as powerful tools for extracting topological information. Both high-pass and low-pass filters are different forms of graph filters that leverage the graph's topology. Therefore, the GNN-favored features selected by TFI remain effective even when applied to high-pass filters. That said, the main focus of this paper is to introduce a novel question and provide a solution for selecting GNN-favored and GNN-disfavored features for graph representation learning. We believe it would be interesting to explore different forms of TFI using advanced graph filters to further enhance the performance of GFS. For instance, as shown in Figure 13, we demonstrate how a high-order neighbor-based graph filter can influence TFI's performance in GFS. Thank you again for your valuable question.

---

> ### Author Response · Authors · 2024-11-21
> **Part(2/2)**
>
> ### W5: 1) A significant concern is the applicability of the proposed GFS. The paper asserts that TFI is computed on training nodes. Thus, I am concerned that the unusual dataset division of 50/25/25 for training/validation/testing employed in this paper is crucial for the effectiveness of GFS. 2) The question then becomes: how would the results be affected if a public splitting, such as 20 per class for training in GCN, SGC, GAT, and APPNP, were applied?
>
> 1. In Figure 6 of our submission, we analyze how the ratio of the training set affects the performance of TFI in GFS. The results indicate that even if using a low training ratio in TFI can sometimes reduce the performance of GFS, it still consistently outperforms the baseline GNNs.
>
> 2. We further conduct the experiments on these datasets with public splits as shown in Table 15. Accuracy results on node classification are reported for Cora, PubMed, and CiteSeer, each having one split. For the other datasets, which have ten splits, we provide both accuracy and standard deviation. The results demonstrate that GNN+GFS outperforms baseline GNN in most datasets for 4 types of GNN backbones, which include GCN, GAT, SGC, and GraphSAGE. This demonstrates the effectiveness of our proposed TFI even under conditions of low supervision.
>
> | **Model**       | **Cora**     | **Pubmed**   | **Citeseer** | **Actor**       | **Chameleon**   | **Cornell**    | **Squirrel**   | **Texas**      | **Wisconsin**  |
> |------------------|--------------|--------------|---------------|------------------|------------------|-----------------|-----------------|----------------|----------------|
> | GCN              | 81.28        | 78.06        | 70.60         | 33.68±0.64       | 62.41±1.97       | 60.81±6.53      | 48.41±1.52      | 66.49±6.14     | 77.65±4.91     |
> | GCN+GFS          | 81.94        | 78.46        | 71.74         | 35.30±0.85       | 62.68±2.02       | 68.11±6.47      | 49.13±1.78      | 77.57±6.50     | 80.39±4.89     |
> | $\Delta$         | **+0.66**    | **+0.40**    | **+1.14**     | **+1.62**        | **+0.27**        | **+7.30**       | **+0.72**       | **+11.08**     | **+2.74**      |
> | GAT              | 80.16        | 76.92        | 69.88         | 33.07±0.79       | 65.79±2.55       | 63.24±4.97      | 51.63±1.47      | 74.05±4.97     | 76.47±3.70     |
> | GAT+GFS          | 81.06        | 77.94        | 72.24         | 35.23±0.83       | 66.49±2.11       | 69.19±5.87      | 52.15±1.83      | 77.84±4.38     | 81.57±6.21     |
> | $\Delta$         | **+0.90**    | **+1.02**    | **+2.36**     | **+2.16**        | **+0.70**        | **+5.95**       | **+0.52**       | **+3.79**      | **+5.10**      |
> | SGC              | 80.74        | 76.96        | 69.84         | 30.64±0.75       | 54.39±1.79       | 43.24±7.64      | 39.91±1.56      | 54.86±4.42     | 53.73±5.64     |
> | SGC+GFS          | 81.48        | 76.38        | 71.18         | 35.62±1.20       | 54.63±2.14       | 67.30±4.84      | 42.19±1.12      | 71.62±4.27     | 78.63±6.43     |
> | $\Delta$         | **+0.74**    | -0.58        | **+1.34**     | **+4.98**        | **+0.24**        | **+24.06**      | **+2.28**       | **+16.76**     | **+24.90**     |
> | SAGE             | 81.06        | 76.80        | 68.52         | 36.20±0.81       | 58.42±2.53       | 70.00±7.03      | 39.32±2.06      | 74.59±5.73     | 81.18±2.11     |
> | SAGE+GFS         | 81.88        | 78.40        | 71.46        | 36.47±0.72       | 58.71±1.93       | 71.08±5.70      | 39.28±1.49      | 81.08±5.55     | 83.14±5.08     |
> | $\Delta$         | **+0.82**    | **+1.60**    | **+2.94**     | **+0.27**        | **+0.29**        | **+1.08**       | -0.04           | **+6.49**      | **+1.96**      |
>
> [1] Feature Selection: A Data Perspective. ACM Comput. Surv. 50(6).
>
> [2] Multi-label Feature Selection via Global Relevance and Redundancy Optimization. IJCAI 2020.
>
> [3] Di Jin, Rui Wang, Meng Ge, Dongxiao He, Xiang Li, Wei Lin, and Weixiong Zhang. Raw-gnn: Random walk aggregation based graph neural network. arXiv preprint arXiv:2206.13953, 2022.
>
> [4] Liang Yang, Mengzhe Li, Liyang Liu, Chuan Wang, Xiaochun Cao, Yuanfang Guo, et al. Diverse message passing for attribute with heterophily. Advances in Neural Information Processing Systems, 34:4751–4763, 2021b.
>
> [5] Yuhan Chen, Yihong Luo, Jing Tang, Liang Yang, Siya Qiu, Chuan Wang, and Xiaochun Cao. Lsgnn: towards general graph neural network in node classification by local similarity. arXiv preprint arXiv:2305.04225, 2023.

---

> > ### Comment · Reviewer_jpAB · 2024-11-25
> >
> > Thanks for the authors’ response, particularly for providing significant experimental results. I have additional concerns regarding the experimental results. Given that Cora, Citeseer, and Pubmed typically achieve scores of around 81.5, 71.0, and 79.0, respectively, on the mentioned split, which is higher than your reported results. Can you explain what led to the results?

---

> ### Author Response · Authors · 2024-11-25
> **Additional Results**
>
> Thank you for your response. **Could you please specify which paper you are referring to regarding these results?** We adhere to the default settings in [1], where only one split is used for experiments. Our findings are comparable to those reported in that paper with 20 labels per class. However, **another approach, referred to as the "public splits," is discussed in [2], which utilizes splits of 60%/20%/20% across 10 runs.** This setting has also been widely used for "public splits" in recent studies [3,4,5,6,7]. These results are based on different settings, which are more aligned with the results you mentioned.
>
> **Could you kindly indicate which paper contains the results you mentioned?** Thank you!
>
> [1] Zhilin Yang, William W. Cohen, Ruslan Salakhutdinov: Revisiting Semi-Supervised Learning with Graph Embeddings. CoRR abs/1603.08861 (2016)
>
> [2] Hongbin Pei, Bingzhe Wei, Kevin Chen-Chuan Chang, Yu Lei, Bo Yang: Geom-GCN: Geometric Graph Convolutional Networks. ICLR 2020
>
> [3] Xiang Li, Renyu Zhu, Yao Cheng, Caihua Shan, Siqiang Luo, Dongsheng Li, Weining Qian: Finding Global Homophily in Graph Neural Networks When Meeting Heterophily. ICML 2022: 13242-13256
>
> [4] Sitao Luan, Chenqing Hua, Qincheng Lu, Jiaqi Zhu, Mingde Zhao, Shuyuan Zhang, Xiao-Wen Chang, Doina Precup: Revisiting Heterophily For Graph Neural Networks. NeurIPS 2022
>
> [5] Yunchong Song, Chenghu Zhou, Xinbing Wang, Zhouhan Lin: Ordered GNN: Ordering Message Passing to Deal with Heterophily and Over-smoothing. ICLR 2023
>
> [6] Kun Wang, Guibin Zhang, Xinnan Zhang, Junfeng Fang, Xun Wu, Guohao Li, Shirui Pan, Wei Huang, Yuxuan Liang: The Heterophilic Snowflake Hypothesis: Training and Empowering GNNs for Heterophilic Graphs. KDD 2024: 3164-3175
>
> [7] Jeongwhan Choi, Seoyoung Hong, Noseong Park, Sung-Bae Cho: GREAD: Graph Neural Reaction-Diffusion Networks. ICML 2023: 5722-5747

---

> > ### Comment · Reviewer_jpAB · 2024-11-25
> >
> > I have pointed out this splitting in Weaknesses 5, as in “How would the results be affected if a public splitting, such as **20 per class for training** in **GCN, SGC, GAT, and APPNP**, were applied?". The results are from these papers. This setup is pivotal for evaluating the applicability of the proposed method, as mentioned in Weaknesses 5.

---

> > > ### Author Response · Authors · 2024-11-25
> > >
> > > We updated our results of **Part(2/2)** using the same setup as in [1], with 20 labels per class for training. We conduct a thorough search for the optimal hyperparameters for the GCN and achieve comparable accuracy in node classification, as you suggested. Our results indicate that GNN combined with GFS outperforms other GNN baselines, demonstrating the effectiveness of GFS in this context.
> > >
> > > Additionally, we would like to highlight that GFS also proves effective across the 10 datasets presented in Table 1, utilizing the splits that are commonly employed in current GNN research [2, 3, 4, 5]. We encourage you to run our code, which is available in the supplementary material or via the anonymous link: https://anonymous.4open.science/r/graph-feature-selection-BF28, to verify the effectiveness of GFS. Thank you.
> > >
> > > **Reference**
> > >
> > > [1] Thomas N Kipf and Max Welling. Semi-supervised classification with graph convolutional networks. arXiv preprint rXiv:1609.02907, 2016.
> > >
> > > [2] Sitao Luan, Chenqing Hua, Qincheng Lu, Jiaqi Zhu, Mingde Zhao, Shuyuan Zhang, Xiao-Wen Chang, Doina Precup: Revisiting Heterophily For Graph Neural Networks. NeurIPS 2022
> > >
> > > [3] Yunchong Song, Chenghu Zhou, Xinbing Wang, Zhouhan Lin: Ordered GNN: Ordering Message Passing to Deal with Heterophily and Over-smoothing. ICLR 2023
> > >
> > > [4] Kun Wang, Guibin Zhang, Xinnan Zhang, Junfeng Fang, Xun Wu, Guohao Li, Shirui Pan, Wei Huang, Yuxuan Liang: The Heterophilic Snowflake Hypothesis: Training and Empowering GNNs for Heterophilic Graphs. KDD 2024: 3164-3175
> > >
> > > [5] Jeongwhan Choi, Seoyoung Hong, Noseong Park, Sung-Bae Cho: GREAD: Graph Neural Reaction-Diffusion Networks. ICML 2023: 5722-5747

---

> ### Author Response · Authors · 2024-11-28
>
> Please let me know if our responses have addressed your concerns. Thank you!

---

> > ### Comment · Reviewer_jpAB · 2024-11-29
> >
> > I carefully read the authors' responses and the revised manuscript. The experimental concerns have been addressed. Therefore, I would like to raise the rating from 3 to 5. However, I am still unclear about the novelty of this paper. Specifically, in the newly proposed metrics and theoretical analysis, the simple replacement of $X$ for the traditional feature selection with $X=A^kX$ for GNNs does not provide many insights.

---

> ### Author Response · Authors · 2024-11-30
> **Reply to your remianing concern on novelty**
>
> We really appreciate your time for the detailed review and raising your score. To address your remaining concern about the novelty of our paper, we will elaborate the importance of our study on $AX$, i.e. the aggregated node features, for graph learning.
>
> Since the main difference between GNN and traditional Neural Networks (NN) is the additional feature aggregation step, investigating how does the aggregation operation influence the model performance stands out as one of the most important topics to study GNN behavior. Recently, considerable efforts have been devoted to this direction and obtained some remarkable insights. For example,
> - [1,6] investigate the structures of the aggregated node embeddings $AX$ under varying levels of homophily, finding that a consistent neighborhood distribution of intra-class nodes enhances GCN performance;
> - [2] studies the node similarities of $AX$ from post aggregation perspective, which results in a new homophily metrics and a proof of the effectiveness of high-pass filter on heterophily problem;
> - [3] studies the impact of graph convolution by comparing the node distinguishability of $X$ and $AX$ under different homophily levels, discovering the mid-homophily pitfall phenomenon;
> - [4] provides a detailed analysis of $AX$ in different homophily intervals and explain the performance disparity of GNN and MLP with graph structural shifts.
>
> While these studies are important, they miss the fine-grained analysis on each feature dimension and do not examine how graph convolution influences GNNs feature-wisely. Our paper fills this gap by providing a feature-wise analysis of the aggregated features, making a valuable contribution to the community.
>
> Besides, in contrast to traditional feature selection methods [5], which aim to reduce redundancy, select informative features, or generate new features, our proposed Graph Feature Selection (GFS) framework has an additional property, that is to evaluate $\textit{"when is a given feature beneficial for graph convolution?"}$. This approach effectively leverages the strengths of both GNNs and non-GNN models. Our experiments demonstrate its effectiveness across eight GNN backbones on various datasets.
>
> [1] Is Homophily a Necessity for Graph Neural Networks?. In International Conference on Learning Representations.
>
> [2] Revisiting heterophily for graph neural networks. Advances in neural information processing systems. 2022 Dec 6;35:1362-75.
>
> [3] When Do Graph Neural Networks Help with Node Classification? Investigating the Homophily Principle on Node Distinguishability. Advances in Neural Information Processing Systems. 2024 Feb 13;36.
>
> [4] Demystifying structural disparity in graph neural networks: Can one size fit all?. Advances in neural information processing systems. 2024 Feb 13;36.
>
> [5] Feature Selection: A Data Perspective. ACM Comput. Surv. 50(6).
>
> [6] Understanding Heterophily for Graph Neural Networks. The Twelfth International Conference on Learning Representations.

---

> ### Author Response · Authors · 2024-12-03
> **Last Minute Discussion**
>
> Dear Reviewer jpAB,
>
> Thanks for all your previous efforts that try to help us improve our paper. We would like to know if your remaining concern on novelty is addressed. We can take this last chance to answer your question. Thanks again.
>
> Best,
>
> Authors

---

### Author Response · Authors · 2024-11-25

We would like to extend our sincere gratitude to all the reviewers for their valuable feedback. In response to the question, "Is Graph Convolution Always Beneficial for Every Feature?", we introduce a new metric, TFI, designed to assess whether specific features benefit from graph-aware models. This is supported by both our empirical observations and theoretical justifications. Following the principle of "feeding the right feature to the right model," we propose GFS, a method that directs GNN-favored features to graph-aware models and GNN-disfavored features to graph-agnostic models. Our experiments demonstrate that this approach can be easily integrated into various GNN backbones, **enhancing model performance without the need of hyperparameter tuning.**

Below is a summary of our additional findings from the rebuttal:

1. **Impact of Label Homophily:** We utilize the CSBM to manipulate label homophily in synthetic graphs. As shown in Table 16, GFS consistently outperforms both MLP and GCN across varying levels of label homophily.

2. **Additional Settings:**
(1) By varying the number of node features in synthetic datasets, we find that GFS is significantly more effective in datasets with a higher number of features, demonstrating greater fault tolerance during feature selection.
(2) Introducing sparsity in node features shows that GFS maintains effectiveness across different levels of sparsity.
(3) When random noises are added to real-world datasets, GFS continuously outperforms baseline GNNs on noisy graphs.

3. **Statistical Analysis of Selected Features:** As illustrated in Table 21, we found that:
(1) Datasets encoded by Pretrained Language Models (PLMs) exhibit similar feature values; notably, a higher average TFI across features correlates with improved GFS performance.
(2) GFS is less effective on datasets with higher sparsity, such as Minesweeper and Tolokers.
(3) Datasets with lower homophily tend to classify more features as GNN-disfavored, while those with higher homophily identify more features as GNN-favored.

Thank you once again to all the reviewers. **We look forward to your response.**

---

### Meta-Review · Area_Chair_a86c · 2024-12-24

**Metareview:**

This paper proposes a novel technique called e Graph Feature Selection (GFS) to enhance the performance of GCNs. The method leverages a Topological Feature Informativeness (TFI) to distinguish GNN-favored features and GNN-disfavored features. The proposed method is justified by theories and experiments.

The paper is well written. The motivation is clearly presented and the results nicely supports the hypothesis that the authors made. The theoretical analysis is solid and detailed. The numerical experiments well support that the proposed method enhances performances of GCNs. Thus, I recommend acceptance of the paper.

**Additional Comments On Reviewer Discussion:**

The reviewers raised some concerns about the experimental settings. However, the authors addressed the concerns by re-conducted the experiments with a common experimental setting. Other detailed technical details were also well resolved during the discussion phase.

---

### Decision · Program_Chairs · 2025-01-22

Accept (Poster)